# Impairment-based assessments for patients with lateral ankle sprain: A systematic review of measurement properties

**Alexander Philipp Schurz**[1,2]*, **Jente Wagemans**[1,3], **Chris Bleakley**[4], **Kevin Kuppens**[3], **Dirk Vissers**[1,3], **Jan Taeymans**[1,2]

**1** Department of Health Professions, Bern University of Applied Sciences, Bern, Switzerland, **2** Faculty of Physical Education and Physiotherapy, Vrije Universiteit Brussel, Brussels, Brussels-Capital Region, Belgium, **3** Department of Rehabilitation Sciences and Physiotherapy, Faculty of Medicine and Health Sciences, University of Antwerp, Antwerp, Belgium, **4** School of Health Science, Ulster University, Newtownabbey, Northern Ireland

* alexander.schurz@bfh.ch, alex-schurz@web.de

## Abstract

### Study design

Systematic review.

### Background and objective

The International Ankle Consortium developed a core outcome set for the assessment of impairments in patients with lateral ankle sprain (LAS) without consideration of measurement properties (MP). Therefore, the aim of this study is to investigate MPs of assessments for the evaluation of individuals with a history of LAS.

### Methods

This systematic review of measurement properties follows PRISMA and COSMIN guidelines. Databases Pubmed, CINAHL, Embase, Web of Science, Cochrane Library and SPORTDiscus were searched for eligible studies (last search: July 2022). Studies on MP of specific tests and patient-reported outcome measurements (PROMs) in patients with acute and history of LAS (>4 weeks post injury) were deemed eligible.

### Results

Ten studies of acute LAS and 39 studies of history of LAS patients with a total of 3313 participants met the inclusion criteria. Anterior Drawer Test (ADT) in supine position five days post injury and Reverse Anterolateral Drawer Test are recommended in acute settings in single studies. In the history of LAS patients, Cumberland Ankle Instability Tool (CAIT) (4 studies) as a PROM, Multiple Hop (3 studies) and Star Excursion Balance Tests (SEBT) (3 studies) for dynamic postural balance testing showed good MPs. No studies investigated pain, physical activity level and gait. Only single studies reported on swelling, range of

**Data Availability Statement:** All relevant data are within the paper and its Supporting Information files.

**Funding:** The authors received no specific funding for this work.

**Competing interests:** The authors (Alexander Philipp Schurz, Jente Wagemans, Chris Bleakley, Kevin Kuppens, Dirk Vissers, Jan Taeymans) have declared that no competing interests exist.

motion, strength, arthrokinematics, and static postural balance. Limited data existed on responsiveness of the tests in both subgroups.

## Conclusion

There was good evidence to support the use of CAIT as PROM, Multiple Hop, and SEBT for dynamic postural balance testing. Insufficient evidence exists in relation to test responsiveness, especially in the acute situation. Future research should assess MPs of assessments of other impairments associated with LAS.

## Introduction

The ankle is the most frequently injured body part in a variety of sports, accounting for 34.30 percent of all injuries. Among these, lateral ankle sprain (LAS) is the most common injury (76.70%) [1]. Swenson et al. [2] identified most injuries to the anterior talofibular ligament (85.30%), followed by the calcaneofibular ligament (34.50%) and the anterior inferior talofibular ligament (26.40%). LAS occur almost twice as often in females as in males (13.60 vs. 6.94 per 1000 exposures). Children are also more susceptible to an ankle sprain [3]. In addition, the risk is increased in indoor and court sports [3]. During matches, LAS occur most frequently in football, with 11.68 cases per 1000 person-hours [1]. Costs relating to the treatment of such injuries ranged from $292 to $2268 per patient (2016 US Dollar) [4].

Thirty to 70 percent of the individuals with a history of an ankle sprain develop a clinical condition defined as chronic ankle instability [5–7]. This is characterized by structural injuries, reduced health-related quality of life, and a range of impairments [8] including pain [9, 10], swelling, decreased ankle range of motion [9], and reduced functionality [11].

Core outcome set (COS) are highly relevant for clinical trials. With its existence clinical conditions can be objectively assessed, and with specified outcomes. The International Ankle Consortium (IAC) has partially addressed this goal of developing a COS for acute LAS patients. This developed Rehabilitation-Oriented Assessment tool (ROAST) [12] highlighted specific tests and patient-reported outcome measurements (PROM) for assessing an ankle joint, after possible fracture or syndesmosis injury are ruled out. The content of ROAST was based primarily on expert consensus, and the measurement properties (MP) of its constituent tests and PROMs, have not been systematically evaluated. Accurate clinical assessment is underpinned by selecting clinical tests and PROMs that have optimal MPs (such as reliability, validity, responsiveness, and interpretability). To assess the quality of such tests, a guideline for the development of systematic reviews was established by the Consensus-based Standards for the selection of health Measurement Instruments (COSMIN) [13–15]. This guideline enables a systematic and transparent approach to the selection of specific tests or PROMs. The aim of this study was to determine whether the COS developed by the IAC for impairment-based assessment is consistent with current evidence specifically for lateral ankle sprains. The specific research question is: Which clinical assessments for the diagnosis of impairments of the ankle joint in individuals with a history of a lateral ankle sprain have the best measurement properties?

This will summarize the current state of research in this field and can highlight where evidence for specific impairment-based tests is lacking. The review can also streamline how clinicians evaluate LAS, by selecting tests with the best MPs. From a research perspective, this study is of value as it presents the current state of research on the assessments for this

population. This knowledge helps to emphasize ideas for future studies, for example, on the development and validation of specific impairment-based tests.

## Methods

Details of the study were registered in the Prospective Register of Systematic Reviews (PROS-PERO) on March 23[rd], 2021 (registration number: CRD42021232513). This systematic review followed the COSMIN guidelines for conducting a systematic review of MPs and the Preferred Reporting Items for Systematic Reviews and Meta-Analyses (PRISMA) guideline for reporting of the research process [13–15].

### Eligibility criteria

Studies were included based on predefined eligibility criteria. Patients were adults between 18 and 65 years of age with acute LAS and history of LAS. An acute ankle sprain was defined as an acute traumatic injury to the lateral ligamentous complex of the ankle joint resulting from excessive inversion of the hindfoot or combined plantar flexion and internal rotation of the foot [16]. A time limit was set at four weeks post injury to be included in the acute population. All ankle sprains occurring more than four weeks before inclusion were assigned to the group of patients defined as history of LAS group. This included both recurrent sprain and CAI patients. A recurrent sprain was defined as two or more sprains at the same ankle joint. CAI was defined as a condition characterized by recurrent sprains of the ankle, the feeling of "giving way" and perceived instability with the initial sprain happening at least twelve months ago [16]. For study populations containing different injuries, at least 70 to 75 percent of participants had to present with LAS to be included. No restrictions were imposed on participants' sex or their physical activity levels.

Included studies must have investigated at least one MP of a clinical test or PROM. If the outcome of interest was not investigated, the study was ineligible for inclusion in this systematic review. Exclusively technical and measurement reports were included. No restrictions were imposed on publication time and language of the studies. A detailed list of inclusion and exclusion criteria is attached (Table 1).

### Information sources and search strategy

Studies were retrieved from a systematic literature search of six electronic databases (Pubmed, CINAHL, Embase, Web of Science, Cochrane Library and SPORTDiscus). Search terms for population, intervention, and outcome of interest were combined with Boolean operators and checked for additional value in the PubMed database (S1 Table). The developed search strategy was then adapted for the other databases mentioned above. Studies were searched from their inception through July 1[st] 2022. No additional grey literature was included. Authors were contacted in case full texts could not be gathered.

### Selection process

The records retrieved during the searches were independently screened based on pre-defined inclusion criteria by two review team members (AS, JW) from January 2021 until March 2021. The study selection was repeated in July 2022. A third reviewer (CB) was contacted in case of disagreement to achieve consensus. The screening was first carried out based on title and abstract and secondly with full texts of the studies. For this process Rayyan QCRI, the systematic reviews web app, was used [17].

**Table 1. Inclusion and exclusion criteria.**

| Criteria | Participants | Age | Intervention | Primary outcomes | Secondary outcomes | Effect size | Study type |
|---|---|---|---|---|---|---|---|
| **Inclusion** | acute lateral ankle sprains recurrent lateral ankle sprains chronic ankle instability | adults (18–65 years) no restrictions on sex or physical activity level | clinical assessments questionnaires | ankle joint: pain swelling range of motion muscle strength arthrokinematics PROMs | static postural balance dynamic postural balance gait physical activity level | Reliability (internal consistency, reliability, measurement error) Validity (content validity, construct validity, criterion validity, structural validity, hypotheses testing) Responsiveness interpretability diagnostic accuracy | technical/ measurement reports |
| **Exclusion** | other diagnosis | other anatomical location than ankle patients with severe ocular impairment any neurologic, cardial, vascular, metabolic diseases | intervention studies | other | other | / | intervention studies other study designs |

## Data collection process

The process of data extraction was similar for the PROMs and clinical tests. In both cases, two team members (AS, JW) independently extracted data and a third reviewer (KK) reassessed the studies in case of discrepancies. The data on general characteristics, study sample, interpretability, and results of studies on measurement characteristics were extracted in an Excel spreadsheet using the COSMIN template [13–15]. Usability of the Excel spreadsheet was independently pilot tested with two included studies by two review team members (AS, JW) and adapted if necessary.

## Data items

The constructs were selected according to the IAC's COS, known as the ROAST guideline [12]. These included a total of ten different outcomes which were separated a priori into primary and secondary outcomes. The assessments of ankle joint pain, swelling, range of motion, muscle strength, arthrokinematics and PROMs were defined as primary outcomes.

Arthrokinematics included assessment of both the extent of movement and the end feeling. Secondary outcomes included static and dynamic postural balance, gait, and physical activity level. Number of subjects, gender distribution, age, diagnosis, and time since injury, if available, were extracted. In addition, the examination position was indicated if it was relevant. The measured construct was extracted for PROM studies. Only those MPs that were examined in the respective study were included in the table. All others were omitted for the sake of clarity.

## Study risk of bias assessment

Studies were assessed by different risk of bias tools, based on the nature of the investigated test or outcome. This assessment was applied at outcome level. Risk of bias assessment was undertaken independently by two research team members (AS, JW). The COSMIN Risk of Bias scale was used for PROM and clinical test studies, with the adapted-COSMIN Risk of Bias used for clinical trials that focused on reliability and measurement error. Diagnostic accuracy studies were assessed with the QUADAS-2 tool. All other clinical studies were also evaluated

with the COSMIN Risk of Bias tool for PROMs. In some cases, two instruments were selected to assess the risk of bias to evaluate all investigated MPs of this study. A consensus meeting was organized with a third researcher (KK) in case of disagreement. The data were entered in an Excel spreadsheet developed according to the COSMIN recommendations. At the beginning, a test run with two studies was carried out and the original spreadsheet was adapted to improve its usability. Data according to the existing consensus of the reviewers were reported per assessment tool.

The COSMIN Risk of Bias tool included the following MPs: reliability (internal consistency, reliability, measurement error), validity (content validity, construct validity, criterion validity), responsiveness, and interpretability [15]. Each category contains sub-questions, which were rated as "very good", "adequate", "doubtful", "inadequate" or "not applicable". The final score was per MP with the worst score counting.

Similar evaluation criteria existed for the adapted-COSMIN tool and its two categories reliability and MP. The evaluation here ranged from "very good", "adequate", "doubtful" to "inadequate" and ultimately also resulted in an overall score per category.

Diagnostic accuracy studies were assessed using QUADAS-2 as recommended by the Agency for Health Care Research and Quality, Cochrane Collaboration [18, 19]. These studies and their measures were selected as diagnostic accuracy studies according to Simundic's criteria [20]. The risk of bias was rated as "low", "high", and "unclear" for diagnostic accuracy studies in the four categories "patient selection", "index test", "reference standard", and "flow of timing". In addition, the areas "patient selection", "index test", and "reference standard" were evaluated in the "applicability concerns" category. Data from the risk of bias analysis were considered during data collection.

## Effect measures

Researchers reached consensus on the taxonomy, terminology and definitions of MPs [21]. These MPs covered four different areas: reliability, validity, responsiveness, and interpretability. Reliability included internal consistency, dependability, and measurement error. Content, construct and criterion validity were defined as validity subcategories. Cross-cultural validity was excluded from this systematic review, as the main aim of this study was to summarize the most valid and reliable instruments for measuring the ten different outcomes. For this reason, only the original version of the questionnaires was used. Minimal important change, minimal important difference (MID), and minimal detectable change (MDC) were assigned by COSMIN to the interpretability section [21].

For the interpretation of the effect sizes, the categories described by COSMIN for good MPs were used. Each result for a MP was subsequently rated as "sufficient" (+), "insufficient" (-), and "indeterminate" (?). This was done independently by two resear1chers (AS, JW). CB was consulted in case of discrepancies. If most of the results were above the limit, the result was rated "sufficient". For reliability, intraclass correlation or weighted kappa values of at least 0.70 were considered "sufficient". Values below this score were categorized as "insufficient". If the intraclass correlation or weighted kappa value was not assessed, this was counted as an "indeterminate" result. For structural validity criteria, "sufficient" rating was either a confirmatory factor analysis using comparative fit index, Tucker-Lewis index or comparable measure of higher than 0.95, a Root Mean Square Error of Approximation of lower than 0.06 or a Standardized Root Mean Residual score of lower than 0.08. Alternatively, the criteria for RASH analysis included no violation of one-dimensionality, local independence, monotonicity and adequate model fit. Internal consistency was defined as low evidence for "sufficient" structural validity and Cronbach´s alpha of at least 0.70 for each unidimensional scale or subscale. If the

intraclass correlation coefficient or weighted Kappa were at least 0.70, the reliability outcome was rated "sufficient". The smallest detectable change or limit of agreement should be lower than the minimal important change for the "sufficient" rating of measurement error. "Sufficient" value for hypothesis testing included a result which was in accordance with the hypothesis. If a correlation with gold standard or an area under the curve value was higher than 0.70, criterion validity was "sufficient". The responsiveness needed to be in accordance with the hypothesis or the area under the curve value was at least 0.70 for a "sufficient" rating. In case criteria were not met, no hypothesis was defined or most of the results were below the limit, the MPs were rated "insufficient". If no hypothesis was explicitly described in the study, defined by the research team or the criteria for a sufficient rating were not mentioned, the result was rated as "indeterminate" [15].

## Synthesis methods and certainty assessment

The criteria from the COSMIN guideline for PROM studies were used for synthesis. Reporting of studies was based on time since injury with two subcategories for each of the ten outcomes: One group included acute LAS, which were defined in the period up to four weeks since injury. The second group compromised all individuals with injury history of more than four weeks since initial LAS including recurrent injuries and CAI were eligible. As data were inconsistent and therefore no pooling of the data was possible, the GRADE approach for certainty assessment could not be carried out [15]. The data was analyzed independently by AS and JW. CB was listened to in case of disagreement.

## Results

### Study selection

The literature search in six databases yielded 5605 studies. Full-text screening led to 49 studies which met the inclusion criteria. Most frequent exclusion reasons in the second screening were wrong population (53%), cross-cultural studies (23%), and wrong outcome (14%). Two studies were not accessible; hence the authors were contacted. No author responded and therefore both studies were excluded from further analysis (Fig 1). A list of excluded studies during full text screening is added (S2 Table).

### Study characteristics

The included forty-nine studies investigated a total of 3313 participants (50% male, 50% female). The study size ranged from eleven to 223 participants. Ten studies included participants with acute ankle sprain. The remaining 39 were conducted on participants with LAS at least four weeks prior the assessment. Of the included studies, 16 studies examined PROMs whereas 33 investigated clinical tests.

### Risk of bias studies

A total of sixteen studies on PROMs and twelve clinical tests were assessed with COSMIN Risk of Bias whereas fifteen clinical test studies were evaluated with the adapted-COSMIN tool. In nine studies, QUADAS-2 was also used. Of the studies with COSMIN Risk of Bias, 62 percent were rated "very good", followed by "adequate" (16%), "inadequate" (16%), and "doubtful" (6%). Using the adapted-COSMIN tool, 37 and 30 percent of respective assessments were rated as adequate or doubtful (very good 6%, inadequate 27%). Most of the criteria in the QUADAS-2 tool were assessed with low risk of bias (72%, high 18%, unclear 10%). A detailed

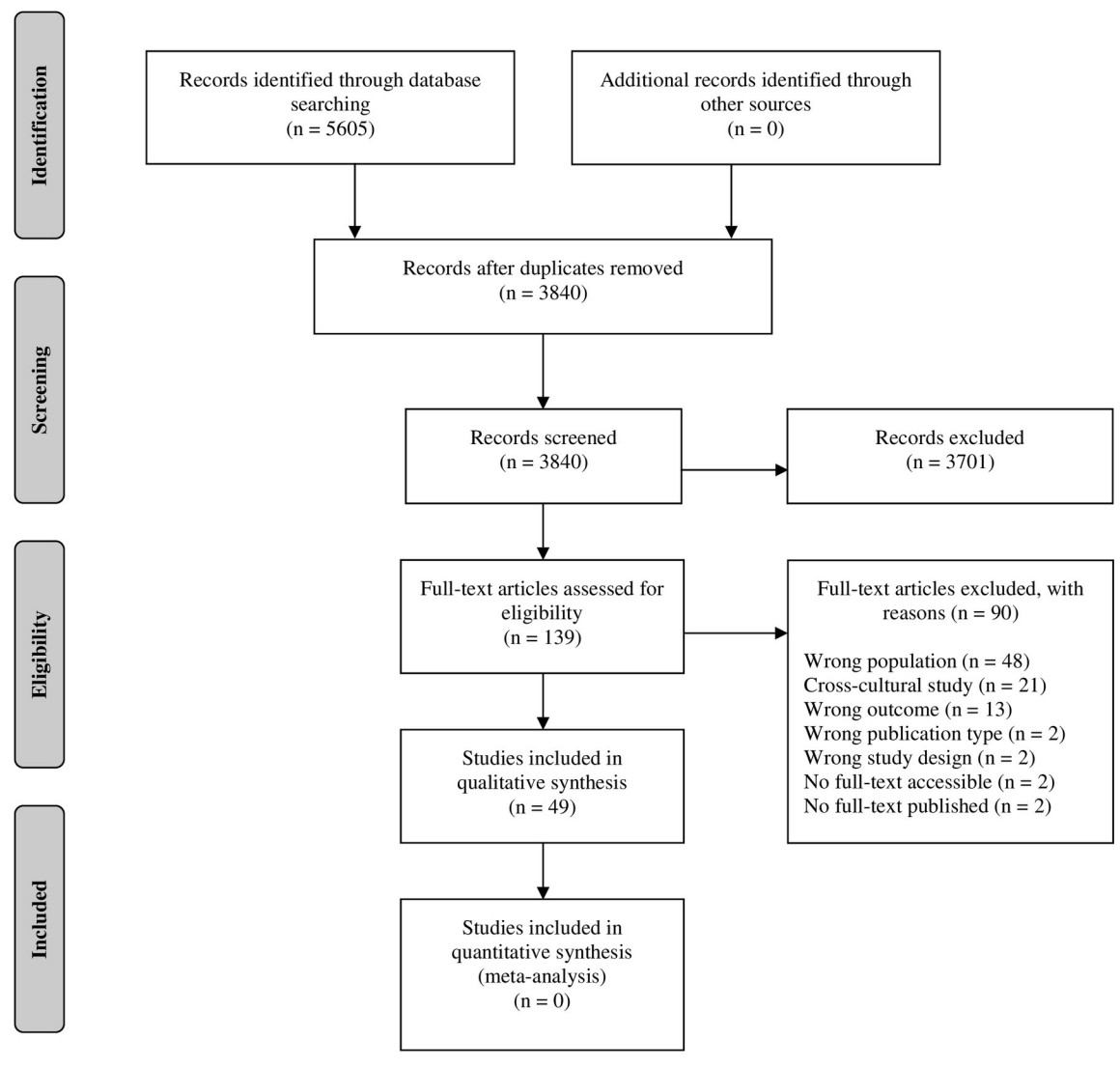

**Fig 1. Flow chart.**

list of the ratings for the risk of bias, broken down by assessment tool, can be found in the appendix (Tables 2 and 3).

## Results of individual studies

Tables 4 and 5 depict all constructs with criteria of good MPs. Diagnostic accuracy studies are not included.

**Ankle joint pain.** No studies existed on the assessment of pain in individuals with LAS.

**Ankle joint swelling.** The Figure of Eight measurement appeared to be reliable (ICC: 0.99) and valid (Criterion validity: r: 0.90) for ankle edema secondary to ankle injuries (acute, recurrent ankle sprain or ankle fracture) [22]. Edema was assessed two days to 29 months between injury and date of measurement. As most of the injuries were assessed within four months since injury, this was allocated to the acute lateral ankle sprain population (Table 6).

**Ankle joint range of motion.** Studies existed exclusively on the history of LAS patients. The Upper and Lower Twist tests had sufficient intra-rater reliability (Kappa scores: 0.70 to

**Table 2. Risk of bias rating COSMIN.**

| Study | Structural validity | Internal consistency | Reliability | Measurement error | Criterion validity | Hypotheses testing | Responsiveness |
|---|---|---|---|---|---|---|---|
| *COSMIN Risk of Bias-PROM studies* | | | | | | | |
| Wright [6] | | | | ia | | | |
| Van der Wees [23] | | | | | vg | ia | with other outcome: ia between subgroups: a |
| Williams [24] | | | ia | | | vg | vg |
| Rosen [25] | | | a | vg | | vg | |
| Wright [26] | | | | | vg | | criterion/ construct: vg |
| Hoch [27] | | | | | | vg | |
| Hoch [28] | | | ia | d | | | criterion: ia; construct: d |
| Hoch [29] | | vg | | | vg | vg | |
| Docherty [30] | | vg | vg | vg | | | |
| Eechaute [31] | | vg | a | vg | | vg | |
| Hale [32] | | | ia | | | vg | construct/ construct: vg |
| Wikstrom [33] | | | | | | vg | |
| Carcia [34] | | | | | | vg | |
| Goulart Neto [35] | | | | | | vg | |
| Hiller [36] | a | | a | | a | | vg |
| Donahue [37] | | a | | | vg | vg | |
| *COSMIN Risk of Bias-Clinical test studies* | | | | | | | |
| Jamsandekar [38] | | | | | | | vg |
| Spahn [39] | | | | | vg | | |
| Lohrer [40] | | | | | vg | vg | |
| Lin [41] | | | | | vg | vg | |
| Wenning [65] | | | | | | other: d, subgroup: vg | |
| Linens [42] | | | | | | vg | vg |
| Yoon [68] | | | ia | ia | ia | | |
| Eechaute [43] | | | | | | | criterion: vg; construct: vg |
| Bastien [44] | | | | | vg | vg | |
| Bolt [45] | | | a | a | | vg | vg |
| Jaffri [46] | | | | d | | | |
| Han [69] | | | a | | | vg | vg |
| **Adapted-COSMIN Risk of Bias-Clinical test studies (assessing only reliability, measurement error)** | | | | | | | |
| Mawdsley [22] | | | d | d | | | |
| Erichsen [47] | | | vg | | | | |
| De Noronha [48] | | | d | | | | |
| Hosseinian [64] | | | ia | | | | |
| Nauck [50] | | | d | | | | |
| Wilkin [51] | | | a | vg | | | |
| Lin [52] | | | d | | | | |
| Laessoe [53] | | | ia | | | | |
| Abdo [49] | | | a | a | | | |
| Lee [67] | | | ia | | | | |
| Eechaute [54] | | | a | a | | | |
| Eechaute [55] | | | a | a | | | |

*(Continued)*

**Table 2.** (Continued)

| Study | Structural validity | Internal consistency | Reliability | Measurement error | Criterion validity | Hypotheses testing | Responsiveness |
|---|---|---|---|---|---|---|---|
| **Pierobon** [56] | | | d | d | | | |
| **Bolt** [45] | | | ia | ia | | | |
| **Jaffri** [46] | | | a | a | | | |

**Note**: **abbreviations**: a = adequate; d = doubtful; ia = inadequate; vg = very good

1.00) in individuals with ongoing symptoms more than three months after ankle distortion, but inter-rater reliability was insufficient (Kappa scores: 0.35 to 0.48) [47]. Reliability for the Plantarflexion and Dorsiflexion tests was insufficient (Kappa scores intra-rater: 0.27 to 0.70) or not calculated [47]. The novel ankle haptic interface system by Lin and colleagues [41] could be used to validly measure mobility (r: 0.99). Sensitivity and specificity values for Weight bearing lunge test, Ankle Plantarflexion, Inversion and Eversion were between 37.7 and 77.4 percent [38] (Table 6).

**Ankle strength.** Testing ankle strength was evaluated only in the history of LAS population. The Baseline digital push-pull dynamometer for plantarflexion, dorsiflexion, inversion and eversion testing, showed specificity values between 50.7 and 86.8 percent. Sensitivity values varied from 60.4 to 81.1 percent [38]. The isokinetic dynamometer could be used to reliably measure force for different angular velocities (120°/s, 30°/s) of inversion or eversion movement in individuals with a chronic condition following LAS (ICC: 0.71 to 0.95) [48] (Table 6).

**Arthrokinematics.** In an acute setting, a delayed clinical examination about five days after the event showed improved sensitivity (acute: 71%; delayed: 96%) and specificity (acute: 33%, delayed: 84%) of the Anterior Drawer Test (ADT) in lying position [57–59]. However, when examined in seated position sensitivity (5 to 93%) and specificity values (67 to 100%) of ADT varied strongly [59–61]. Spahn [39] found a good correlation (r: 0.91) of clinical examination with the ADT in lying examination position and the results of stress radiography. Stress sonography was still the most specific (specificity 87 to 100%) method for detecting injuries to the anterior talofibular ligament of the ankle joint compared to arthrometer and clinical examination [59, 60]. Sonography showed insufficient values for inter-rater reliability (Kappa values: 0.158 to 0.640) but was especially specific for the diagnostic testing (Specificity values: 76 to 99%) [64]. Moderate sensitivity values (44.70 to 50%) and high specificity values (97 to

**Table 3. Risk of bias rating QUADAS-2.**

| QUADAS-2 Risk of Bias-Clinical test studies | | | | | Applicability concerns | | |
|---|---|---|---|---|---|---|---|
| Study | Patient Selection | Index Test | Reference Standard | Flow and Timing | Patient Selection | Index Test | Reference Standard |
| **van Dijk** [57] | low | low | low | low | low | low | low |
| **van Dijk** [58] | low | low | low | low | low | low | low |
| **Wiebking** [59] | high | low | low | unclear | low | low | unclear |
| **George** [60] | low | low | low | low | low | low | low |
| **Li** [61] | high | low | low | low | low | low | low |
| **Hosseinian** [64] | low | low | low | low | low | low | low |
| **Gomes** [62] | high | low | high | low | low | low | low |
| **Chen** [66] | low | unclear | unclear | low | low | unclear | unclear |
| **Rosen** [63] | high | low | high | low | low | low | high |

**Table 4. Overview table of study results: Primary outcomes.**

| Clinical tests | Study | Structural validity | Internal consistency | Reliability | Measurement error | Criterion validity | Hypotheses testing | Responsiveness |
|---|---|---|---|---|---|---|---|---|
| *Ankle Pain* | | | | | | | | |
| *Ankle Swelling-lateral ankle sprains ≤4 weeks prior* | | | | | | | | |
| Figure of Eight | Mawdsley [22] | | | + | ? | | | |
| *Ankle Joint Range of Motion-lateral ankle sprain >4 weeks prior* | | | | | | | | |
| Upper Twist Test | Erichsen [47] | | | + (intra) - (inter) | | | | |
| Lower Twist Test | Erichsen [47] | | | + (intra) - (inter) | | | | |
| Plantarflexion Test | Erichsen [47] | | | - (inter) - (inter) | | | | |
| Dorsiflexion Test | Erichsen [47] | | | -/ nc | | | | |
| Novel Ankle Range of Motion | Lin [41] | | | | | + | ? | |
| Weight bearing lunge test | Jamsandekar [38] | | | | | | | ? |
| Ankle Plantarflexion–supine position | Jamsandekar [38] | | | | | | | ? |
| Inversion–hook lying | Jamsandekar [38] | | | | | | | ? |
| Eversion–hook lying | Jamsandekar [38] | | | | | | | ? |
| *Ankle Strength-lateral ankle sprain >4 weeks prior* | | | | | | | | |
| Baseline digital Push-pull Dynamometer—Plantarflexion | Jamsandekar [38] | | | | | | | ? |
| Baseline digital Push-pull Dynamometer—Dorsiflexion | Jamsandekar [38] | | | | | | | ? |
| Baseline digital Push-pull Dynamometer—Inversion | Jamsandekar [38] | | | | | | | ? |
| Baseline digital Push-pull Dynamometer—Eversion | Jamsandekar [38] | | | | | | | ? |
| Isokinetic Dynamometer | De Noronha [48] | | | + | | | | |
| *Arthrokinematics-lateral ankle sprain ≤4 weeks prior* | | | | | | | | |
| Anterior Drawer Test | Spahn [39] | | | | | + | | |
| Sonography | Hosseinian [64] | | | - (inter) | | | | |
| *Arthrokinematics-lateral ankle sprain >4 weeks prior* | | | | | | | | |
| Ankle Arthrometer | Nauck [50] | | | + | | | | |
| | Lohrer [40] | | | | | - | ? | |
| Anterior Drawer Test supine | Wilkin [51] | | | inter: - | ? | | | |
| Anterior Drawer Test crook lying | Wilkin [51] | | | inter: - | ? | | | |
| Anterior Drawer Test seated | Lin [52] | | | + | | | | |
| Anterior Drawer Test | Wenning [65] | | | | | | ? | |
| Talar Tilt Test | Wilkin [51] | | | inter: - | ? | | | |
| Inversion Tilt Test | Wilkin [51] | | | inter: - | ? | | | |
| Wireless Sonography (Sonostar Technologies Co.) | Wenning [65] | | | | | | ? | |
| *PROM-lateral ankle sprain >4 weeks prior* | | | | | | | | |
| Ankle Function Scale | Van der Wees [23] | | | | | + | ? | ? |

(*Continued*)

**Table 4.** (Continued)

| Clinical tests | Study | Structural validity | Internal consistency | Reliability | Measurement error | Criterion validity | Hypotheses testing | Responsiveness |
|---|---|---|---|---|---|---|---|---|
| **Sport Ankle Rating System** | Williams [24] | | | QoL: + | | | ? | ? |
| *PROM-lateral ankle sprain ≤4 weeks prior* | | | | | | | | |
| **Ankle Instability Instrument (AII)** | Docherty [30] | | ? | + | ? | | | |
| **Chronic Ankle Instability Scale (CAIS)** | Eechaute [31] | | ? | + | ? | | ? | |
| **Cumberland Ankle Instability Tool (CAIT)** | Wright [6] Rosen [25] Wright [26] Hiller [36] | ? | | +, + | ? | +, +, - | ?,? | |
| **Foot and Ankle Ability Measure (FAAM)** | Carcia [34] | | | | | | ? | |
| | Goulart Neto [35] | | | | | | ? | |
| **Quick-FAAM (electronic)** | Hoch [27] | | | | | | ? | |
| **Quick-FAAM** | Hoch [28] Hoch [29] | | ? | + | ? | ? | ? | ? |
| **Foot and Ankle Disability Index (FADI)** | Hale [32] Wikstrom [33] | | | + | | | ?,? | + |
| **FADI-Sport** | Hale [32] Wikstrom [33] | | | + | | | ?,? | + |
| **Identification of Functional Ankle Instability (IDFAI)** | Donahue [37] | | ? | + | ? | | ? | |
| **Foot and Ankle Outcome Score (FAOS)** | Goulart Neto [35] | | | | | | ? | |

*Note*: *abbreviations*: FAAM = Foot and Ankle Ability Measure; FADI = Foot and Ankle Disability Index; inter = interrater; intra = intrarater; nc = not calculated;

PROM = Patient-reported outcome measurement

results of the criteria of good measurement property: +: sufficient, -: insufficient,?: indeterminate

table adapted from COSMIN recommendations, Diagnostic accuracy studies are not mentioned as grading of measurement properties was not possible with these study

100%) were observed for the Anterolateral Drawer Test [61]. In order to diagnose chronic anterior talofibular ligament injuries, the Reverse Anterolateral Drawer Test seemed to be more sensitive (86.80% to 92.10%) and accurate than the ADT or Anterolateral Drawer Test (κ values: ADT: 0.20, ALDT: 0.53, RALDT: 0.64) [61]. Talar Tilt Test (TTT) showed reasonable sensitivity (54%) for first line screening [60].

In the population of more than four weeks of LAS history, the ankle arthrometer appeared to be a reliable tool for objective detection of anterior talar drawer instability (ICC: 0.80) [50]. The study by Lohrer and colleagues showed low correlation of the ankle arthrometer values with the FAAM (Foot and Ankle Ability Measure) results and thus low criterion validity (Activity of Daily Living subscale: r: 0.29, Sport subscale: r: 0.32) [40]. Wilkin assessed poor inter-rater reliability values for manual ADT in supine (ICC: 0.16 to 0.23) and crook lying position (ICC: 0.06 to -0.12) [51]. Contrasting results concerning reliability (ICC > 0.90) were observed for the ADT in seated position by Lin [52]. In CAI patients, the ADT seemed to be especially specific (100%) but less sensitive (50%) and accurate (69.60%) for instability testing [62]. ADT showed indeterminate results for hypotheses testing in comparison with Cumberland Ankle Instability Tool (rho: -0.81) or Forgotten Joint Score (rho: -0.75) [65]. Using an instrumented ADT sensitivity (80.4%) and specificity (86.3%) results were high [66]. The manual TTT showed insufficient values for reliability (ICC: 0.22 to 0.33), indeterminate values for

**Table 5. Overview table of study results.** Secondary outcomes.

| Clinical tests | Study | Structural validity | Internal consistency | Reliability | Measurement error | Criterion validity | Hypotheses testing | Responsiveness |
|---|---|---|---|---|---|---|---|---|
| *Static Postural Balance-lateral ankle sprain >4 weeks prior* | | | | | | | | |
| **Balance Error Scoring System** | Linens [42] | ? | | | | | ? | ? |
| **Time in Balance Test** | Linens [42] | ? | | | | | ? | ? |
| **Foot Lift Test** | Linens [42] | ? | | | | | ? | ? |
| **Centre of Pressure** | Linens [42] | ? | | | | | ? | ? |
| **Instrumented wobble board** | Laessoe [53] | | | + | | | | |
| **Digital assessment "My Ankle"** | Abdo [49] | | | - | ? | +, - | ? | |
| **Single Leg Heel Raise Balance Test** | Lee [67] | | | + | | | | |
| **Smartphone Accelerometer** | Yoon [68] | | | + | ? | - | | |
| *Dynamic Postural Balance-lateral ankle sprain >4 weeks prior* | | | | | | | | |
| **Multiple Hop Test** | Eechaute [54] Eechaute [55] Eechaute [43] | | | +, + | ?,? | - | ?,? | ? |
| **Star Excursion Balance Test** | Linens [42] Pierobon [56] Bastien [44] | ? | | + | ? | + | ?,? | ? |
| **Side Hop Test** | Linens [42] | ? | | | | | ? | ? |
| **Figure of Eight Hop Test** | Linens [42] | ? | | | | | ? | ? |
| **Step Down Test** | Bolt [45] | | | - | ? | | ? | ? |
| **Dynamic Leap and Balance Test** | Jaffri [46] | ? | | + | ? | | | |
| **Ankle Inversion Discrimination Apparatus for Landing (AIDAL)** | Han [69] | | | + | | | ? | |
| *Gait* | | | | | | | | |
| *Physical Activity Level* | | | | | | | | |

*Note*: results of the criteria of good measurement property: +: sufficient, -: insufficient,?: indeterminate

table adapted from COSMIN recommendations, Diagnostic accuracy studies are not mentioned as grading of measurement properties was not possible with these study

hypotheses testing (comparing with Cumberland Ankle Instability Tool: rho: -0.83, comparing Forgotten Joint Score: rho: -0.78), but seemed to be specific (78 to 88%) with lower values for sensitivity (49%) in CAI evaluations [51, 63, 65]. Using an arthrometer for the Inversion Tilt Test led to low sensitivity (36%) but moderate to high specificity values (72 to 94%) [63]. Manual Inversion Tilt Testing yielded insufficient values for reliability in ankle sprain and other subjects with ankle injuries (ICC: 0.26 to 0.29) [51]. The Anterior Talar Palpation test is highly sensitive (sensitivity 100%, specificity 77.80%) with good diagnostic accuracy (91.30%) in the detection of ankle instability [62]. Indeterminate results were found for hypotheses testing of stress sonography using a wireless sonography (compared to Cumberland Ankle Instability Tool: rho: -0.44 to -0.48, to Forgotten Joint Score: -0.35 to -0.41) [65] (Table 6).

**Ankle joint specific patient-reported outcome measurements.** Two PROMs were found in the literature for the evaluation of acute ankle sprains. Ankle Function Score (AFS) has criterion validity for evaluating function (AFS & Olerud-Molander Ankle Score: r: 0.70–0.82) [23]. Using the Sport Ankle Rating System and its three subscales (Quality of Life, Clinical

**Table 6. Summary of findings.** Primary outcomes.

| Outcome measure | Study | Patients | Diagnosis | Measurement property | Result |
|---|---|---|---|---|---|
| *Ankle swelling-lateral ankle sprain ≤4 weeks prior* | | | | | |
| **Figure of Eight** | Mawsdley [22] | n = 15 7–8 (47%-53%) 22.7y, SD: 4.42 | ankle sprain or musculoskeletal injury, 2d-41m | reliability | ICC: 0.99 (+) |
| | | | | reliability: measurement error | SEM: first: 0.44cm (?); second: 0.44cm (?); third: 0.45 (?) |
| *Ankle joint range of motion-lateral ankle sprain >4 weeks prior* | | | | | |
| **Upper Twist Test** | Erichsen [47] | n = 27 16–11 (59%-41%) range: 20-45y | ankle distortion; ≥3m | reliability: intra-rater reliability reliability: inter-rater reliability | Kappa scores e1: 0.87 (+), e2: 0.72 (+) Kappa scores: e1: 0.48 (-), e2: 0.35 (-) |
| **Lower Twist Test** | Erichsen [47] | n = 27 16–11 (59%-41%) range: 20-45y | ankle distortion; ≥3m | reliability: intra-rater reliability reliability: inter-rater reliability | Kappa scores: e1: 1.00 (+), e2: 0.70 (+) Kappa scores: e1: 0.37 (-), e2: 0.37 (-) |
| **Plantarflexion Test** | Erichsen [47] | n = 24 17–7 (69%-31%) range: 20-45y | ankle distortion; ≥3m | reliability: intra-rater reliability reliability: inter-rater reliability | Kappa scores: e1: aPF: 0.47 (-); pPF 0.61 (-); e2: aPF: 0.70 (+); pPF 0.43 (-) Kappa scores: e1: aPF: 0.35 (-); pPF 0.20 (-); e2: aPF: 0.49 (-); pPF 0.47 (-) |
| **Dorsiflexion Test** | Erichsen [47] | n = 24 17–7 (69%-31%) range: 20-45y | ankle distortion; ≥3m | reliability: intra-rater reliability reliability: inter-rater reliability | Kappa scores: e1: aDF: nc; pDF: nc; e2: aDF: 0.61 (-); pDF: 0.27 (-) Kappa scores: e1: aDF: 0.17 (-); pDF: nc; e2: aDF: nc; pDF: nc |
| **Novel Ankle Range of Motion measurement** | Lin [41] | n = 19 - (%-%) h: 24.7y, SD: 1.9 p: 24.4y, SD: 1.3 | ankle instability | validity: criterion validity (concurrent) | PF/DF sagittal plane: r: 0.99 (+) IV/EV in frontal plane: r: 0.99 (+) |
| | | | | validity: hypotheses testing for construct approach | ROM for 37 angular positions in sagittal/ frontal plane between 0.70°: r: 0.99 (?) |
| **Weight bearing lunge test** | Jamsandekar [38] | n = 106 69–36 (65%-35%) h: 21.0y, SD: 2.1 p: 21.9y, SD: 2.6 | CAI, healthy | responsiveness | specificity: 77.4% (?), sensitivity: 71.7% (?) |
| **Ankle Plantarflexion–supine position** | Jamsandekar [38] | n = 106 69–36 (65%-35%) h: 21.0y, SD: 2.1 p: 21.9y, SD: 2.6 | CAI, healthy | responsiveness | specificity: 64.2% (?), sensitivity: 56.6% (?) |
| **Inversion–hook lying** | Jamsandekar [38] | n = 106 69–36 (65%-35%) h: 21.0y, SD: 2.1 p: 21.9y, SD: 2.6 | CAI, healthy | responsiveness | specificity: 37.7% (?), sensitivity: 54.7% (?) |
| **Eversion–hook lying** | Jamsandekar [38] | n = 106 69–36 (65%-35%) h: 21.0y, SD: 2.1 p: 21.9y, SD: 2.6 | CAI, healthy | responsiveness | specificity: 49.1% (?), sensitivity: 54.7% (?) |
| *Ankle strength-lateral ankle sprain >4 weeks prior* | | | | | |
| **Baseline digital Push-pull Dynamometer—Plantarflexion** | Jamsandekar [38] | n = 106 69–36 (65%-35%) h: 21.0y, SD: 2.1 p: 21.9y, SD: 2.6 | CAI, healthy | responsiveness | specificity: 50.9% (?), sensitivity: 60.4% (?) |
| **Baseline digital Push-pull Dynamometer—Dorsiflexion** | Jamsandekar [38] | n = 106 69–36 (65%-35%) h: 21.0y, SD: 2.1 p: 21.9y, SD: 2.6 | CAI, healthy | responsiveness | specificity: 54.7% (?), sensitivity: 62.3% (?) |

*(Continued)*

**Table 6.** (Continued)

| | | | | | |
|---|---|---|---|---|---|
| **Baseline digital Push-pull Dynamometer—Inversion** | Jamsandekar [38] | n = 106 69–36 (65%-35%) h: 21.0y, SD: 2.1 p: 21.9y, SD: 2.6 | CAI, healthy | responsiveness | specificity: 69.8% (?), sensitivity: 67.9% (?) |
| **Baseline digital Push-pull Dynamometer—Eversion** | Jamsandekar [38] | n = 106 69–36 (65%-35%) h: 21.0y, SD: 2.1 p: 21.9y, SD: 2.6 | CAI, healthy | responsiveness | specificity: 86.8% (?), sensitivity: 81.1% (?) |
| **Isokinetic Dynamometer** | De Noronha [48] | n = 11 11–0 (100%-0%) range: 18-25y | lateral ankle sprain, ≥4m, max: 12m | reliability | ICC: i: 120°/s IV/ EV: 0.89–0.92 (+); 30°/s IV/ EV: 0.71–0.90 (+) ni: 120°/s IV/ EV: 0.92–0.94 (+); 30°/s IV/ EV: 0.90–0.95 (+) |

| Outcome measure | Study | Patients | Diagnosis | Examination position | Measurement property | Result |
|---|---|---|---|---|---|---|
| *Arthrokinematics-lateral ankle sprain ≤4 weeks prior* | | | | | | |
| **Anterior Drawer Test** | Van Dijk [57] | n = 160 116–44 (73%-27%) 27.3y, SD: - range: 18–40 | acute inversion trauma | lying position | diagnostic accuracy | PE < 48 hours: sensitivity: 71%, specificity: 33% delayed PE experienced investigator: sensitivity: 96%, specificity: 84% |
| | Van Dijk [58] | n = 160 116–44 (73%-27%) 27.3y, SD: - range: 18–40 | acute inversion trauma | lying position | diagnostic accuracy | PE <48h: sensitivity: 71%, specificity: 33% physical examination 5 days after injury: sensitivity: 96%, specificity: 84% |
| | Wiebking [59] | n = 30 17-13(57%-43%) 35y, SD: 14 | lateral ankle sprain, anterior talofibular ligament injury | seated position of the patient, 90° flexion of knee | diagnostic accuracy | sensitivity: 93%, specificity: 67% |
| | George [60] | n = 35 17–18 (49%-51%) 21.97y, SD: 7.11 | history of LAS, mean 3.6w (SD: 3.32) since injury | seated position with calf hanging over edge of examination bed | diagnostic accuracy | sensitivity: 59% |
| | Li [61] | n = 31 18–13 (55%-45%) median: h: 29.1y; SD: 8.9 p: 30.4y; SD: 8.9 | lateral ankle sprain, >1w since injury | seated position with calf hanging over edge of examination bed | diagnostic accuracy | first tester: sensitivity: 5.3%; specificity: 100% FNR: 0.947; FPR: 0, LR-: 0.95 second tester: sensitivity: 39.5%, specificity: 100% FNR: 0.605, FPR: 0, LR-: 0.61 |
| | Spahn [39] | n = 16 3–13 (19%-81%) 32.7y, SD: 11.3 | lateral ankle sprain (mean: 14.1h; SD: 12.4, since injury) | lying position, with device | validity: criterion validity | ADT-ADT+ stress sonography: r: 0.91 (+) |
| **Stress Sonography** | Wiebking [59] | n = 30 17-13(57%-43%) 35y, SD: 14 | lateral ankle sprain, anterior talofibular ligament injury | | diagnostic accuracy | stress-sonography device (3mm cut-off value): sensitivity: 27%, specificity: 87%, AUC: 0.51 |
| | George [60] | n = 35 17–18 (49%-51%) 21.97y, SD: 7.11 | history of LAS, mean 3.6w (SD: 3.32) since injury | | diagnostic accuracy | specificity: 100% |

(*Continued*)

**Table 6.** (Continued)

| | | | | | | |
|---|---|---|---|---|---|---|
| **Sonography** | Hosseinian [64] | n = 105<br>47–58 (55%-45%)<br>32.95y; SD: 1.55 | Acute ankle ligament injury | | reliability: inter-rater reliability | Kappa: sprain: 0.158 to 0.640 (-)<br>partial sprain: 0.384 to 0.741 (-),<br>complete tear: 0.670 to 0.840 (+),<br>partial + complete tear: 0.420 to 0.860 (-) |
| | | | | | diagnostic accuracy | sprain: sensitivity: 27% to 88%, specificity: 76% to 97%<br>partial sprain: sensitivity: 33% to 78%, specificity: 76% to 99%<br>complete tear: sensitivity: 80% to 82%, specificity: 86% to 99%<br>partial + complete tear: sensitivity: 33% to 87%, specificity: 98% to 99% |
| **Arthrometer** | Wiebking [59] | n = 30<br>17-13(57%-43%)<br>35y, SD: 14 | lateral ankle sprain, anterior talofibular ligament injury | | diagnostic accuracy | sensitivity: 80%, specificity: 40%, AUC: 0.44 |
| **Antero-lateral Drawer Test** | Li [61] | n = 31<br>18–13 (55%-45%)<br>median: h: 29.1y; SD: 8.9<br>p: 30.4y; SD: 8.9 | lateral ankle sprain, >1w since injury | seated position with calf hanging over edge of examination bed | diagnostic accuracy | first tester: sensitivity: 44.7%, specificity: 100%<br>FNR: 0.553, FPR: 0, LR- 0.55<br>second tester: sensitivity: 50%, specificity: 97.1%<br>FNR: 0.5, FPR: 0; LR+ 17.2, LR-: 0.51 |
| **Reverse Antero-lateral Drawer Test** | Li [61] | n = 31<br>18–13 (55%-45%)<br>median: h: 29.1y; SD: 8.9<br>p: 30.4y; SD: 8.9 | lateral ankle sprain, >1w since injury | lying position | diagnostic accuracy | first tester: sensitivity: 86.8%, specificity: 91.2%<br>FNR: 0.132, FPR: 0.088; LR+: 9.9, LR-: 0.14<br>second tester: sensitivity: 92.1%, specificity: 88.2%<br>FNR: 0.079, FPR: 0.118; LR+: 7.8, LR-: 0.09 |
| **Talar Tilt Test** | George [60] | n = 35<br>17–18 (49%-51%)<br>21.97y, SD: 7.11 | history of lateral ankle sprain, mean 3.6w (SD: 3.32) since injury | seated position | diagnostic accuracy | sensitivity: 54% |

*Arthrokinematics-lateral ankle sprain >4 weeks prior*

| | | | | | | |
|---|---|---|---|---|---|---|
| **Ankle Arthrometer** | Nauck [50] | n = 23<br>- (43%-57%)<br>24y, SD: 12 | history of lateral ankle sprain | seated, hip & knee 90° flexion | reliability | ICC for stiffness analysis: 0.80 (+) |
| | Lohrer [40] | n = 41<br>/<br>h: 26.3y, SD: 4.7<br>p1: 24.9y, SD: 2.3<br>p2: 32.9y, SD: 13.5 | residual symptoms after ankle sprain: CAI patients and me severe CAI patients, >1y | not specified | validity: criterion validity | individual 40-60N arthrometer stiffness values-FAAM-G<br>ADL: r: 0.286 (-), Sport: r: 0.316 (-) |
| | | | | | validity: validity: hypotheses testing for construct approach | FAI subjects-MAI subjects: stiffness 40-60N;<br>p: 0.006 (?),<br>Stiffness 125–175 N; p: 0.468<br>FAI subjects-MAI patients: stiffness 40-60N; p<0.001,<br>stiffness 125-175N; p: 0.773 (?)<br>MAI subjects-MAI patients: stiffness 40-60N;<br>p: 0.224, stiffness 125–175 N; p: 0.844 (?) |

(*Continued*)

**Table 6.** (*Continued*)

| | | | | | | |
|---|---|---|---|---|---|---|
| **Anterior Drawer Test** | Wilkin [51] | n = 60<br>- (15%-85%)<br>range: 17–50 | ankle sprain, no-ankle sprain,<br>other conditions: ligament reconstructive surgery, previous malleolar fracture and congenital metatarsus varus | supine | reliability: interrater reliability | ICC between 3 raters: 0.16 (-)<br>ICC between the experienced raters: 0.23 (-) |
| | | | | | reliability: measurement error | SEM: between 3 raters: 1.11 (?) between the experienced raters: 1.19 (?) |
| | | | | crook lying | reliability: interrater reliability: | ICC between 3 raters: 0.06 (-)<br>ICC between the experienced raters: -0.12 (-) |
| | | | | | reliability: measurement error | SEM: between 3 raters: 1.39 (?)<br>SEM: between the experienced raters: 1.69 (?) |
| | Lin [52] | n = 16<br>- (69%-31%)<br>h: 24.6y, SD: 2.3<br>p: 24.8y, SD: 1.8 | lateral ankle sprain, within 1 year since injury | seated, hip & knee 90° flexion | reliability | ICC > 0.90 (+) |
| | Gomes [62] | n = 24<br>- (64%-36%)<br>28y, range: 23–42 | ankle instability, other complaints, 18.3m since injury | seated or lying supine | diagnostic accuracy | sensitivity: 50%; specificity: 100%; PPV: 100%; NPV: 56.3%<br>kappa: 0.44; accuracy: 69.6% |
| | Wenning [65] | n = 50<br>not stated<br>h: 23.6y, SD: 4.0<br>MAI: 24.6y, SD: 4.7 | mechanical ankle instability | not stated | validity: hypotheses testing for construct approach | physical examination & CAIT: rho: -0.81 (?)<br>physical examination & Forgotten Joint Score: rho: -0.75 (?) |
| **Instrumented ADT** | Chen [66] | n = 313<br>156–157 (49%-51%)<br>h: 30.49y, SD: 8.11<br>CAI: 30.84y, SD: 9.43 | CAI, healthy | not specified | diagnostic accuracy | sensitivity: 80.4%; specificity: 86.3% |
| **Talar Tilt Test** | Rosen [63] | n = 88<br>- (49%-51%)<br>h: 20.2y, SD: 1.3<br>CAI: 20.7y, SD: 1.5<br>p: 22.1y, SD: 4.1 | CAI, copers, healthy | not specified | diagnostic accuracy | sensitivity: 49%; specificity: 78% to 88%; LR+: 2.23 to 4.14;<br>LR-: 0.58–0.66 |
| | Wilkin [51] | n = 60<br>- (15%-85%)<br>range: 17–50 | ankle sprain, no-ankle Sprain,<br>other conditions: ligament reconstructive surgery, previous malleolar fracture and congenital metatarsus varus | supine, knee flexed to 90° | reliability: interrater reliability | ICC between 3 raters: 0.33 (-)<br>ICC between the experienced raters: 0.22 (-) |
| | | | | | reliability: measurement error | SEM: between 3 raters: 0.93 (?)<br>SEM: between the experienced raters: 1.06 (?) |
| | Wenning [65] | n = 50<br>not stated<br>h: 23.6y, SD: 4.0<br>MAI: 24.6y, SD: 4.7 | mechanical ankle instability | not stated | validity: hypotheses testing for construct approach | physical examination & CAIT: rho: -0.83 (?)<br>physical examination & Forgotten Joint Score: rho: -0.78 (?) |

(*Continued*)

**Table 6.** (Continued)

| | | | | | | |
|---|---|---|---|---|---|---|
| **Inversion Tilt Test with Arthrometer** | Rosen [63] | n = 88<br>- (49%-51%)<br>h: 20.2y, SD: 1.3<br>CAI: 20.7y, SD: 1.5<br>p: 22.1y, SD: 4.1 | CAI, copers, healthy | supine with the test leg extended, the knee flexed to approximately 15˚ | diagnostic accuracy | arthrometer inversion talar tilt test: sensitivity: 36%; specificity: 72%-94%; LR+: 1.26–6.10; LR-: 0.68–0.89 |
| **Inversion Tilt Test-Manual** | Wilkin [51] | n = 60<br>- (15%-85%)<br>range: 17–50 | ankle sprain, no-ankle sprain,<br>other conditions: ligament reconstructive surgery, previous malleolar fracture and congenital metatarsus varus | supine, knee flexed to 90˚ | reliability: interrater reliability | ICC between 3 raters: 0.29 (-)<br>ICC between the experienced raters: 0.26 (-) |
| | | | | | reliability: measurement error | SEM: between 3 raters: 0.98 (?)<br>SEM between the experienced raters: 1.04 (?) |
| **Anterior Talar Palpation** | Gomes [62] | n = 24<br>- (64%-36%)<br>28y, range: 23–42 | ankle instability, other complaints, 18.3m since injury | supine | diagnostic accuracy | Sensitivity: 100%; specificity: 77,8%; PPV: 87,5%; NPV: 100% kappa: 0.81; accuracy: 91.3% |
| **Wireless Sonography (Sonostar Technologies Co.)** | Wenning [65] | n = 50<br>not stated<br>h: 23.6y, SD: 4.0<br>MAI: 24.6y, SD: 4.7 | mechanical ankle instability | not stated | validity: hypotheses testing for construct approach | stress sonography & CAIT-Score: ADT: rho: -0.48 (?), TTT: -0.44 (?)<br>stress sonography & Forgotten Joint Score: ADT: rho: -0.35 (?), TTT: -0.41 (?) |

| Outcome measure | Study | Patients | Diagnosis | Construct | Measurement property | Result |
|---|---|---|---|---|---|---|
| *PROM-lateral ankle sprain ≤4 weeks prior* | | | | | | |
| **Ankle Function Score** | Van der Wees [23] | n = 107<br>65–42 (61%-39%)<br>32y, SD: 14.1 | acute ankle injury (8.7d since injury) | evaluation for recovery after acute ankle injury | validity: criterion validity | AFS-OMAS: r: 0.70–0.82 (+)<br>AFS-PSC: rho: 0.26-0.49 (-) |
| | | | | | validity: hypotheses testing for construct approach | sensitivity at baseline for prognosis of recovery at 2 weeks after injury: 76%, specificity: 63%, predictive value (will not recover within 2 weeks): 86% (?)<br>predictive value (light injury + will recover within 2 weeks): 45% (?) |
| | | | | | responsiveness | ES: 2.00 (?); SRM: 2.10 (?) |
| **Sport Ankle Rating System** | Williams [24] | n = 30<br>26–4 (83%-17%)<br>19.7y, SD: 1.1 | lateral ankle sprain grade 2 | impact on function and psychosocial status of ankle sprain patients | reliability | Quality of life: Cronbach´s alpha: 0.87–0.89 (+) |
| | | | | | validity: hypotheses testing for construct approach | SANE, quality of life, Clinical Rating Scale:<br>lateral ankle sprain vs. healthy group scores: p<0.001 (?) |
| | | | | | responsiveness | Quality of life (all subscales): 0.98–2.93 (?)<br>Clinical rating scale (all subscales): 0.08–5.14 (?) |
| *PROM-lateral ankle sprain >4 weeks prior* | | | | | | |
| **Ankle Instability Instrument (AII)** | Docherty [30] | n = 101<br>29–72 (29%-71%)<br>20.7y, SD: 2.7 | history of ankle sprain | FAI | internal consistency | Cronbach´s alpha overall: 0.89 (?) |
| | | | | | reliability | ICC overall: 0.95 (+) |
| | | | | | measurement error | SEM: 1.85 (?) |

*(Continued)*

**Table 6.** (Continued)

| | | | | | | |
|---|---|---|---|---|---|---|
| **Chronic Ankle Instability Scale (CAIS)** | Eechaute [31] | n = 29 / 25y, SD: 5.4 | CAI, 76.3 months disease duration | quantification of multidimensional profile of CAI | reliability: internal consistency | Cronbach´s alpha (for each subscale): impairment: 0.62–0.64 (?); disability: 0.71–0.80 (?); participation: 0.68–0.74 (?); emotions: 0.62–0.74 (?) |
| | | | | | reliability | ICC total: 0.84 (+) |
| | | | | | reliability: measurement error | SEM: 2.7 (?) |
| | | | | | validity: hypotheses testing for construct approach | rho: with talar tilt values: 0.05–0.07 (?) with timed performance of multiple hop test: 0.38–0.40 (?) VAS score of multiple hop test: 0.41–0.49 (?) |
| | | | | | interpretability | MDC: 4.7 |
| **Cumberland Ankle Instability Tool (CAIT)** | Wright [6] | n = 50 12–38 (24%-76%) 21.5y, SD: 4.4 | CAI, 7.04y disease duration | CAI | responsiveness | largest Youden index 0.893 (?); ≤23 ideal cutoff to distinguish group membership in this dataset Youden index for cutoff ≤25: 0.834 (?), with ≤25: sensitivity: 96.6%; specificity: 86,6%; LR-: 0.039, LR+: 7.318 recalibration: largest Youden index: 0.95 (?) indicated CAIT ≤25 as ideal cutoff, sensitivity of cutoff: 95.1%, specificity of cutoff: 100% LR-: 0.049, LR+: not calculated, nearest LR+: 27.171 |
| | | | | | interpretability | MDC: 3.08 MCID ≥3 |
| | Rosen [25] | n = 68 37–31 (46%-54%) m: 27.3y, SD: 7.6 f: 22.9y, SD: 4.9 | history of ankle sprain | CAI | reliability | test-retest: ICC: 0.86 (+) |
| | | | | | reliability: measurement error | SEM CAIT-paper: 0.78 (?) CAIT-digital T1: 0,73 (?); CAIT-digital T2: 0.88 (?) |
| | | | | | validity: hypotheses testing for construct approach | ICC: 0.86–0.93(?); weighted Kappa: 0.67–0,81 (?) |
| | | | | | interpretability | MDC: CAIT-paper: 1.41; CAIT-digital: T1 2.02, T2 2.42 |
| | Wright [26] | n = 118 46–72 (39%-61%) h: 25.02y, SD: 5.49 p: 25.52y, SD: 6.31 | CAI | CAI | validity: criterion validity | AUC: 0.988 (+); recalibration: AUC: 0.996 (+) |
| | Hiller [36] | n = 151 concurrent 23y, SD: 6.1 construct 23y, SD: 6.8 discriminative 23y, SD: 6.8 reliability 41y, SD: 9.4 | subjects with no history of ankle sprain, history of unilateral and bilateral sprains | FAI | reliability | ICC: 0.96 (+) |
| | | | | | validity: structural validity | discrimination score for functional ankle instability: 27.5 sensitivity: 82.9% (?); specificity: 74.7% (?); LR+ 3.27 (?), LR- 0.23 (?) |
| | | | | | validity: criterion validity | CAIT-VAS: 0.76 (+); LEFS-CAIT 0.50 (-) |
| | | | | | validity: hypotheses testing for construct approach | optimal discrimination score 27.5, bands: 2.5 points wide LR associated with: being highest band: 0.20 (?) lowest band (<21.5) 32.0 (?) |

(*Continued*)

**Table 6.** (Continued)

| | | | | | | |
|---|---|---|---|---|---|---|
| **Foot and Ankle Ability Measure (FAAM)** | Carcia [34] | n = 30 16–14 (53%-47%) h: 19.8y, SD: 1.0 p: 20.4y, SD: 1.4 | CAI | CAI | validity: hypotheses testing for construct approach | Kendall tau rank correlation ADL subscale-ADL global rating of function: h: r: 0.64 (?), p: r: 0.23 (?) sports subscale & sports global rating of function: r: 0.57 (?) |
| | Goulart Neto [35] | n = 50 29–21 (58%-42%) 27.2y, SD: 6.3 | CAI | Postural control, muscle strength in CAI patients | validity | Postural control: FAAM-Motor Control Test: r: -0.35 (?) FAAM-mSEBT: r: 0.40 (?) FAAM-mBEES: r: -0.26 (?) Muscle strength: FAAM-Invertor: r: 0.42 (?) FAAM-Evertor: r: 0.24 (?) FAAM-Plantar flexors: r: 0.38 (?) FAAM-Dorsiflexor: r: 0.29 (?) FAAM-External hip rotators: r: 0.30 (?) Ankle dorsiflexion: FAAM-Lunge test: r: 0.19 (?) |
| **Quick-FAAM** | Hoch [27] (electronic) | n = 40 13–27 (32%-68%) 23.25y, SD: 4.79 | CAI | CAI | validity: hypotheses testing for construct approach | Quick FAAM-FAAM: r: 0.95 (?) Quick FAAM-GRF-sport: r: 0.71 (?) Quick-FAAM-GRF ADL: r: 0.65 (?) Quick-FAAM-SF-12: r: 0.45 (?) Quick-FAAM-SF-12 physical: r: 0.45 (?) Quick-FAAM-SF-12-mental: r: 0.14 (?) |
| | Hoch [28] | n = 20 / 24.35y, SD: 6.95 | CAI | CAI | reliability | ICC: 0.82 (+) |
| | | | | | reliability: measurement error | SEM: 4.56% (?) |
| | | | | | responsiveness | T2-T3 ES: 1.27–1.49 (?); T2-T4 ES: 1.49 (?) |
| | | | | | interpretability | MDC: 6.5% |
| | Hoch [29] | n = 223 65–158 (29%-71%) h: 24.0y, SD: 5.4 p: 23.4y, SD: 4.5 | CAI | CAI | validity: internal consistency | Cronbach´s Alpha: 0.94 (?) |
| | | | | | validity: criterion validity | Quick-FAAM-CAIT: r: 0.76 (?) |
| | | | | | validity: hypotheses testing for construct approach | Quick FAAM: CAI 74,6% 96.9% (?) CAIT: CAI: 16.4; ASC: 26.8 (?) discriminative capability: sensitivity: 96%; specificity: 85%; AUC: 0.95; cut off score: 94.79% for discriminating between CAI & ASC groups |
| **Foot and Ankle Disability Index (FADI)** | Hale [32] | n = 50 21–29 (42%-58%) 21.53y, SD: 3.59 | CAI | functional limitation in CAI patients | reliability | ICC: 0.89–0.91 (+) |
| | | | | | validity: hypotheses testing for construct approach | rehab group: pre-post intervention: t: 3.29; ES: 0,52 (?) side-by side comparison F: 20.71 p < .0005 |
| | Wikstrom [33] | n = 48 24–24 (50%-50%) h: 20.8y, SD: 1.5 p: 21.7y, SD: 2.8 | CAI | differentiation between CAI patients & copers | validity: hypotheses testing for construct approach | sensitivity at cut off 0.75 (?); 1-specificity at cut off: 0.17 (?); LR+: 4.41 (?); LR-: 0.30 (?) |
| | | | | | responsiveness | AUC: 0.81 (+) |

*(Continued)*

**Table 6.** (Continued)

| | | | | | | |
|---|---|---|---|---|---|---|
| **FADI-Sport** | Hale [32] | n = 50<br>21–29 (42%-58%)<br>21.53y, SD: 3.59 | CAI | functional limitation in CAI patients | reliability | ICC: 0.67–0.84 (+) |
| | | | | | validity: hypotheses testing for construct approach | rehab group: pre-post intervention t: 5,82; ES: 0.71 (?)<br>side-by side comparison F:42.13 |
| | Wikstrom [33] | n = 48<br>24–24 (50%-50%)<br>h: 20.8y, SD: 1.5<br>p: 21.7y, SD: 2.8 | CAI | differentiation between CAI patients & copers | validity: hypotheses testing for construct approach | sensitivity at cut off: 0.67 (?); 1-specificity at cut off: 0.12 (?); LR+: 5.58 (?); LR-: 0.38 (?) |
| | | | | | responsiveness | AUC: 0.79 (+) |
| **Identification of Functional Ankle Instability (IDFAI)** | Donahue [37] | n = 110<br>54–56 (49%-51%)<br>19.80y, SD: 1.4 | history of ankle sprain | FAI | validity: internal consistency | Cronbach´s alpha: overall: 0.96 (?) |
| | | | | | reliability | ICC: 0.92 (+) |
| | | | | | measurement error | SEM: 2.76 (?) |
| | | | | | validity: hypotheses testing for construct approach | IDFAI-LEFS (rho values): overall: -0.38 (?) |
| **Foot and Ankle Outcome Score (FAOS)** | Goulart Neto [35] | n = 50<br>29–21 (58%-42%)<br>27.2y, SD: 6.3 | CAI | Postural control, muscle strength in CAI patients | validity: hypotheses testing for construct approach | Postural control: FAOS-Motor Control Test: r: -0.43 to -0.18 (?)<br>FAOS -mSEBT: r: 0.28 to 0.51 (?)<br>FAOS -mBEES: r: -0.30 to -0.02 (?)<br>Muscle strength: FAOS -Invertor: r: 0.38 to 0.50 (?)<br>FAOS -Evertor: r: 0.20 to 0.33 (?)<br>FAOS -Plantar flexors: r: 0.31 to 0.45 (?)<br>FAOS -Dorsiflexor: r: 0.04 to 0.41 (?)<br>FAOS -External hip rotators: r: 0.16 to 0.31 (?)<br>Ankle dorsiflexion: FAOS -Lunge test: r: 0.1 to 0.22 (?) |

*Note*: *participants*: total amount; male-female (%); mean age, SD.

*abbreviations*: aDF = active Dorsiflexion; ADL = Activity of Daily Living; ADT = Anterior Drawer Test; AFS = Ankle Function Scale; aPF = active Plantarflexion; ASC = Ankle Sprain Copers; CAI = Chronic Ankle Instability; CAIT = Cumberland Ankle Instability Tool; DF = Dorsiflexion; e1 = examiner 1; e2 = examiner 2; ES = Effect Size; EV = Eversion; FAAM = Foot Ankle Ability Measure; FAAM-G = Foot and Ankle Ability Measure-German; FAI = Functional Ankle Instability; FAOS = Foot and Ankle Outcome Score; FNR = False Negative Rate, FPR = False Positive Rate; GRF = Global Rating of Function; h = healthy; i = injured ankle; ICC = Intraclass Correlation Coefficient; IDFAI = Identification of Functional Ankle Instability; IV = Inversion; LEFS = Lower Extremity Functional Scale; LR- = negative Likelihood Ratio; LR+ = positive Likelihood Ratio; MAI = Mechanical Ankle Instability; mBEES = modified Balance Error Scoring System; mSEBT = modified Star Excursion Balance Test; nc = not calculated; ni = noninjured ankle; NPV = negative Predictive Value; OMAS = Olerud-Molander Ankle Score; p = patient (p1, p2); pDF = passive Dorsiflexion; PE = physical examination; PF = Plantarflexion; pPF = passive Plantarflexion; PPV = positive Predictive Value; PROM = Patient-reported outcome measurement; PSC = patient-specific complaints; r = Pearson correlation; rho = Spearman correlation; ROM = Range of Motion; SANE = Sport Ankle Rating System; SEM = standard error of measurement; SF-12 = Short Form 12; SRM = standardized response mean; T2 = timepoint prior to first intervention; T3 = timepoint 24 hours post-intervention; VAS = Visual Analog Scale

results of the criteria of good measurement property are added in brackets: +: sufficient; -: insufficient;?: indeterminate

table adapted from COSMIN recommendations

Rating Score and Single Assessment Numeric Evaluation) the influence of an acute LAS on functional and psychological status could be assessed reliably (Cronbach´s alpha: 0.85 to 0.91) [24].

Thirteen studies examined persons with history of LAS patients. Several PROMs assessed CAI. The Cumberland Ankle Instability Tool (CAIT) was a reliable (ICC: 0.86) and valid (AUC: 0.99, recalculation: 0.99) PROM for evaluation of this construct. Several papers

calculated mean detectable change values (paper version: Rosen 2019: MDC: 1.41, digital version: Wright 2017: MDC: 2.02/2.42, MCID: ≥3) [6, 25, 26]. Additionally, Quick-FAAM was a reliable (ICC: 0.82) questionnaire which was responsive to treatment for the evaluation of CAI (MDC: 6.50%) [27–29]. Criterion validity (r: 0.76) and internal consistency (Cronbach´s alpha: 0.94) were indeterminate [29]. The Ankle Instability Instrument (AII) (ICC: 0.95), Chronic Ankle Instability Scale (CAIS) (ICC: 0.84), Foot and Ankle Disability Inventory (FADI) (ICC: 0.89 to 0.91) and FADI-Sport (ICC: 0.67 to 0.84) were all reliable PROMs for the evaluation of CAI [30–33]. The Foot and Ankle Ability Measure (FAAM) and Foot and Ankle Outcome Score (FAOS) showed indeterminate results for validity for CAI patients [34, 35].

CAIT reliably (ICC: 0.96) and validly (Criterion validity: CAIT and VAS: r: 0.76) assessed functional ankle instability. Criterion validity compared with the Lower Extremity Functional Scale (LEFS) questionnaire was insufficient (CAIT and LEFS: r: 0.50) [36]. An alternative for functional ankle instability evaluation was the Identification of Functional Ankle Instability Scale with proven reliability (ICC: 0.92) [37] (Table 6).

**Static postural balance.** Static postural balance tests were evaluated only in the population with LAS at least four weeks prior. While the Time in Balance Test (AUC: 0.73) and Foot Lift Test (AUC: 0.76) showed sufficient values for hypotheses testing, this was not the case for Balance Error Scoring System (AUC: 0.62) and Centre of Pressure testing (AUC: 0.54 to 0.72) [42]. The Instrumented Wobble Board showed reliable results in the investigation of functional ankle instabilities (ICC: anterior-posterior plane: 0.70, medial-lateral plane: 0.87) [53]. The application "My Ankle" revealed insufficient results concerning its test-retest reliability (ICC: 0.34 to 0.81). Balance in patients with CAI during the closed eye examination was validly assessed with this application (Rho: eyes closed 0.63 to 0.87, eyes opened -0.25 to 0.18) [49]. Single Leg Heel Raise Balance Test showed sufficient test-retest reliability in CAI patients [67]. Intra- (ICC: 0.87 to 0.93) and inter-tester reliability (ICC: 0.82 to 0.90) were sufficient for the Smartphone Accelerometer with insufficient criterion validity (comparison with Cumberland Ankle Instability Tool: 0.33, I-Balance: 0.30) [68] (Table 7).

**Dynamic postural balance.** Several studies existed for the assessment of dynamic postural balance in the history of LAS population. For the detection of functional performance deficits in CAI patients, the Multiple Hop Test seemed to be reliable for test-retest (ICC: patient 0.83 to 0.97), intra- (ICC: 0.94) as well as inter-rater reliability (ICC: 0.94) [43, 54, 55]. All four clinical tests for dynamic postural balance (Star Excursion Balance Test, Side Hop Test, Figure of Eight Hop Test) found an indeterminate result in structural validity and hypotheses testing [42]. Other studies on Star Excursion Balance Test showed its reliability for testing dynamic postural control (ICC: 0.72 to 0.93) and the criterion validity (d > 0.70) [44, 56]. While the Step Down Test showed insufficient results of reliability for the assessment of dynamic postural control (ICC: 0.15 to 0.63) [45], the latter could be reliably assessed using the Dynamic Leap and Balance Test (ICC: 0.85 to 0.96) [46]. The Ankle Inversion Discrimination Apparatus for Landing (AIDAL) has sufficient test-retest reliability (ICC: 0.804) for CAI individuals and indeterminate results for hypotheses testing and responsiveness [69] (Table 7).

**Gait.** The systematic literature search revealed no study on gait for LAS patients.

**Physical activity level.** None of the included studies investigated a test for physical activity level testing in LAS patients.

## Discussion

To the best of our knowledge, this is the first review to summarize current evidence on impairment-based testing and MPs in outcomes used for individuals with a history of ankle sprain. In the acute setting, delayed examination five days after the event with ADT in supine

**Table 7. Summary of findings.** Secondary outcomes.

| Outcome measure | Study | Patients | Diagnosis | Measurement property | Result |
|---|---|---|---|---|---|
| *Static postural balance-lateral ankle sprain >4 weeks prior* | | | | | |
| **Balance Error Scoring System** | Linens [42] | n = 34<br>8–26 (24%-76%)<br>h: 23y, SD: 3<br>p: 23y, SD: 4 | CAI | validity: structural validity | overall, errors: 0.71 (?) |
| | | | | validity: hypotheses testing for construct approach | AUC: 0.62 (?) |
| | | | | responsiveness | cut off scores to identify postural instabilities: single-limb stance on a firm surface: ≥3 (?) errors; total: ≥14 errors (?); sensitivity: 0.47 (?); Youden index: 35.29 (?) |
| **Time in Balance Test** | Linens [42] | n = 34<br>8–26 (24%-76%)<br>h: 23y, SD: 3<br>p: 23y, SD: 4 | CAI | validity: structural validity | 0.92 (?) |
| | | | | validity: hypotheses testing | AUC: 0.73 (?) |
| | | | | responsiveness | cut off scores to identify postural instabilities: ≤25.89 seconds (?); sensitivity: 0.82 (?); Youden index: 47.06 (?) |
| **Foot Lift Test** | Linens [42] | n = 34<br>8–26 (24%-76%)<br>h: 23y, SD: 3<br>p: 23y, SD: 4 | CAI | validity: structural validity | 0.94 (?) |
| | | | | validity: hypotheses testing | AUC: 0.76 (?) |
| | | | | responsiveness | cut off scores to identify postural instabilities: ≥5 lifts (?); sensitivity: 0.76 (?); Youden index: 52.94 (?) |
| **Centre of Pressure** | Linens [42] | n = 34<br>8–26 (24%-76%)<br>h: 23y, SD: 3<br>p: 23y, SD: 4 | CAI | validity: structural validity | anterior-posterior velocity: 0.38 (?); medial-lateral velocity: 0.22 (?); anterior-posterior excursion: 0 (?); medial-lateral excursion: 0.18 (?); anterior-posterior SD: 0.11 (?); medial-lateral SD: 0; rectangular area: 0.001 (?) time to boundary: anterior-posterior mean minima: 0.71 (?); medial-lateral mean minima: 0.30 (?); SD minima: 0.72 (?); anterior-posterior absolute minima: 0.13 (?); SD minima: 0.87 (?); medial-lateral abs minima: 0.11 (?) |
| | | | | validity: hypotheses testing for construct approach | AUC: area: 95% confidence ellipse: 0.56 (?); rectangular area: 0.56 (?); resultant velocity: 0.72 (?) anterior-posterior velocity mean: 0.65 (?); medial-lateral velocity mean: 0.55 (?); anterior-posterior excursion mean: 0.54 (-); SD 0.53; medial-lateral excursion mean: 0.56 (?); SD 0.54 |
| | | | | responsiveness | cut off scores to identify postural instabilities; resultant velocity: ≤1.56 cm/s; anterior-posterior TTB SD: ≤3.78 seconds; medial-lateral TTB SD: ≤1.56 seconds; Youden Index: area 95% confidence ellipse: 29.50; rectangular area: 17.28; resultant velocity: 41.18 anterior-posterior velocity mean: 35.30; medial-lateral velocity mean: 17.60 anterior-posterior excursion mean: 29.40; SD 17.60; medial-lateral excursion mean: 23.60; SD: 29.4 (?) |
| **Instrumented Wobble Board** | Laessoe [53] | n = 50<br>23–27 (46%-54%)<br>h: 24.5y, SD: 2.1<br>p: 25.0y, SD: 4.1 | FAI | reliability | ICC: medio-lateral plane: 0.87 (+); anterior-posterior plane: 0.70 (+) |
| Digital Assessment "My Ankle" | Abdo [49] | n = 67<br>17–50 (25%-75%)<br>median:<br>h: 22y, SD: -<br>range: 19–31<br>p: 22y, SD: -<br>range: 20–27 | CAI, maximum 1year | reliability | ICC: 0.34–0.81 (-) |
| | | | | reliability: measurement error | SEM: 0.22–1.58 (?) |
| | | | | validity: criterion validity | My Ankle and Biodex balance system: eyes-closed testing p: ρ: 0.63–0.87 (+); h: ρ: 0.44–0.62 (-) eyes-open testing: p: ρ: -0.08–0.87 (-); h: ρ: -0.08–0.62 (-) |
| | | | | validity: hypotheses testing for construct approach | negligible ES: 0.03–0.22 (?) |
| | | | | interpretability | MDC: 0.61–4.37 |

*(Continued)*

**Table 7.** (Continued)

| Outcome measure | Study | Patients | Diagnosis | Measurement property | Result |
|---|---|---|---|---|---|
| **Single Leg Heel Raise Balance Test** | Lee [67] | n = 52<br>37–15 (71%-29%)<br>h: 26y, SD: 5.9<br>CAI: 25, SD: 6.9 | CAI, healthy | reliability: test-retest reliability | h: ICC: 0.91 (+), CAI: ICC: 0.87 (+)<br>modified: h: ICC: 0.86 (+), CAI: ICC: 0.80 (+) |
| **Smartphone Accelerometer** | Yoon [68] | n = 26<br>not stated<br>19.82y, SD: 1.60 | CAI | reliability: intra-tester reliability | eyes open: ICC: 0.87–0.90 (+), eyes closed: ICC: 0.90–0.93 (+) |
| | | | | reliability: inter-tester reliability | eyes open: ICC: 0.87–0.90 (+), eyes closed: ICC: 0.82 (+) |
| | | | | reliability: measurement error | intra-tester: eyes open:0.009–0.01 (?), eyes closed: 0.01–0.02 (?)<br>inter-tester: eyes open: 0.01 (?), eyes closed: 0.01 (?) |
| | | | | validity: criterion validity (concurrent) | Accelerometer &CAIT: 0.33 (-)<br>Accelerometer & I-Balance: 0.30 (-) |

*Dynamic postural control-lateral ankle sprain >4 weeks prior*

| Outcome measure | Study | Patients | Diagnosis | Measurement property | Result |
|---|---|---|---|---|---|
| **Multiple Hop Test** | Eechaute [54] | n = 58<br>38–20 (66%-34%)<br>h: 21.8y, SD: 3.4<br>p: 24.9; SD: 5.5 | CAI | reliability | test-retest: ICC time: h: 0.87 both sides (+); p: left: 0.91 (+); right: 0.97 (+)<br>rho: VAS scores patients 0.81 (unstable right ankles) (+); 0.88 (left unstable ankles) (+) |
| | | | | reliability: measurement error | SEM in sec: h: left: 1.9 (?) right: 2.0 (?); p: left: 2.3 (?) right: 2.2 (?) |
| | Eechaute [55] | n = 58<br>38–20 (66%-34%)<br>h: 21.8y, SD: 3.4<br>p: 24.9; SD: 5.5 | CAI | reliability | intra-observer: ICC: h: 0.83 (+); p: 0.94 (+);<br>inter-observer: ICC: h: 0.91 (+); p: 0.94 (+);<br>test-retest reliability: ICC: h: 0.64 (-); p: 0.83 (+) |
| | | | | reliability: measurement error | SEM: h: 2.8 (?); p: 2.6 (?) |
| | | | | validity: criterion validity | correlation of number of balance errors with timed test performance: r: 0.60 (-)<br>with the perceived difficulty of multiple hop test r: 0.4 (-) (p<0.05) |
| | | | | validity: hypotheses testing for construct approach | test (p: 0.000) retest (p: 0.000)<br>based on activity level: competitive athletes: test (p:0.000) retest (p:0.000); recreational: test occasion: p: 0.028; retest occasion: p: 0.014<br>Between feet in CAI patients: test: p: 0.240, retest: p: 0.005; change in support strategy: test: p: 0.000), retest: p: 0.00; fixed support strategy: test: p: 0.173, retest: p: 0.353 (?) |
| | Eechaute [43] | n = 58<br>38–20 (66%-34%)<br>h: 21.8y, SD: 3.4<br>p: 24.9; SD: 5.5 | CAI | validity: hypotheses testing for construct approach | AUC: balance errors: 79% (?); time values: 77% (?); VAS Score: 65% (?)<br>cut off: 13.5 errors; 35 seconds; 32.5 mm (?) |
| | | | | responsiveness | SEM: healthy Errors: 2.8 (?); time value in seconds: 2.1 (?);<br>VAS score millimetre: 9.3 (?); patients errors: 2.6 (?);<br>time value in seconds: 2.3 (?); VAS score in millimetre: 9.9 (?) |
| | | | | interpretability | MDC: p: errors: 7.2; time value in sec: 6.4; VAS score, mm: 27.4 |
| | | | | diagnostic accuracy | 1 of 3 outcomes positive: sensitivity: 100%; specificity: 38%;<br>LR+: 1.6; LR-: 0.00<br>2 of 3 outcomes positive: sensitivity: 86%; specificity: 79%;<br>LR+: 4.2; LR-: 0.17<br>3 outcomes positive: sensitivity: 48%; specificity: 90%;<br>LR+: 4.7; LR-: 0.58 |

(*Continued*)

**Table 7.** (Continued)

| Outcome measure | Study | Patients | Diagnosis | Measurement property | Result |
|---|---|---|---|---|---|
| **Star Excursion Balance Test** | Linens [42] | n = 34<br>8–26 (24%-76%)<br>h: 23y, SD: 3<br>p: 23y, SD: 4 | CAI | validity: structural validity | anteromedial: 0.59 (?); medial: 0.59(?); posteromedial: 0.66 (?) |
| | | | | validity: hypotheses testing for construct approach | AUC: anteromedial reach direction: 0.65 (?); medial: 0.65 (?); posteromedial reach direction: 0.71 (?) |
| | | | | responsiveness | cut off scores to identify postural instabilities: PM ≤0.91 (?);<br>sensitivity: anteromedial: 0.76 (?); medial:0.59 (?); posteromedial: 0.65 (?)<br>Youden index: anteromedial 29.41 (?); medial: 29.41 (?); posteromedial: 35.29 (?) |
| | Bastien [44] | n = 20<br>20–0 (100%-0%)<br>h: 26.0y, SD: 5.1<br>p: 26.2y, SD: 6.9 | lateral ankle sprain | validity: criterion validity | maximal reach distance (% LLL): anteromedial: 7.84%; d: 1.30 (+);<br>medial: 5.11%; d: 0.79 (+); posteromedial: 5.32%; d: 0.81 (+);<br>overall: 6.06%; d: 1.06 (+)<br>maximal reach distance (% height): anteromedial: 8.63%; d: 1.40 (+);<br>medial: 8.97%; d: 0.96 (+); posteromedial: 6.49%; d: 1.09 (+);<br>overall: 7.01%; d: 1.29 (+) |
| | | | | validity: hypotheses testing for construct approach | overall (all direction and groups: 0.991 (?)<br>anteromedial direction for both groups: ICC 0.991 (?)<br>medial direction for both groups: 0.992 (?)<br>posteromedial direction for both groups: 0.990 (?)<br>overall for lateral ankle sprain group: 0.986 (?)<br>overall for healthy group: 0.992 (?) |
| | Pierobon [56] | n = 31<br>12–19 (39%-61%)<br>24y<br>range: 21–30.5 | lateral ankle sprain, <6 weeks since injury | reliability | ICC: anterior: 0.87 (+); anteromedial: 0.87 (+); medial: 0.93 (+);<br>posteromedial: 0.75 (+); posterior: 0.72 (+); posterolateral: 0.78 (+);<br>lateral: 0.87 (+); anterolateral: 0.85 (+) |
| | | | | reliability: measurement error | anterior: 3.09 (?); anteromedial 3.13 (?); medial: 2.43 (?); posteromedial: 4.82 (?); posterior: 4.69 (?); posterolateral: 4.81 (?); lateral: 4.39 (?); anterolateral: 3.50 (?) |
| | | | | interpretability | MDC: anterior: 8.56 anteromedial 8.68; medial 6.73 posteromedial 13.36; posterior: 13.00, posterolateral: 13.33; lateral: 12.17, anterolateral: 9.69 |
| **Side Hop Test** | Linens [42] | n = 34<br>8–26 (24%-76%)<br>h: 23y, SD: 3<br>p: 23y, SD: 4 | CAI | validity: structural validity | 0.65 (?) |
| | | | | validity: hypotheses testing for construct approach | AUC: 0.70 (?) |
| | | | | responsiveness | cut off scores to identify postural instabilities: ≥12.88 sec (?);<br>sensitivity: 0.65 (?); Youden index: 47.06 (?) |
| **Figure of Eight Hop Test** | Linens [42] | n = 34<br>8–26 (24%-76%)<br>h: 23y, SD: 3<br>p: 23y, SD: 4 | CAI | validity: structural validity | 0.49 (?) |
| | | | | validity: hypotheses testing for construct approach | AUC: 0.66 (?) |
| | | | | responsiveness | cut off scores to identify postural instabilities: ≥17.36 seconds (?);<br>sensitivity: 0.47 (?); Youden index: 35.29 (?) |
| **Step Down Test** | Bolt [45] | n = 46<br>11–35 (24%-76%)<br>h: 30.5y, SD: 7.33<br>p: 29.3y, SD: 7.67 | CAI | reliability | ICC: h: forward anterior-posterior: 0.15 (-); forward medial-lateral: 0.62 (-);<br>lateral anterior-posterior: 0.27 (-); lateral medial-lateral: 0.35 (-)<br>p: forward anterior-posterior: 0.12 (-); forward medial-lateral: 0.63 (-); lateral anterior-posterior: 0.30 (-); lateral medial-lateral: 0.33 (-) |
| | | | | reliability: measurement error | SEM: h: forward anterior-posterior: 0.21 (?); forward medial-lateral: 0.11 (?);<br>lateral anterior-posterior: 0.11 (?); lateral medial-lateral: 0.22 (?)<br>p: forward anterior-posterior: 0.33 (?); forward medial-lateral: 0.12 (?);<br>lateral anterior-posterior: 0.11 (?); lateral medial-lateral: 0.22 (?) |
| | | | | validity: hypotheses testing for construct approach | comparison between healthy and CAI: AUC: 0.50–0.57 (?) |
| | | | | responsiveness | SDC: forward: 0.11–0.33s (?); lateral: 0.11–0.22s (?) |

(*Continued*)

**Table 7.** (Continued)

| Outcome measure | Study | Patients | Diagnosis | Measurement property | Result |
|---|---|---|---|---|---|
| **Dynamic Leap and Balance Test** | Jaffri [46] | n = 30 12–18 (40%-60%) h: 19.07y, SD: 0.82 p: 21.06, SD: 3.29 | CAI | validity: structural validity | rater 1: time: ES: 1,97 (?); p: 0,001; errors: ES 1,05 (?); p: 0,004 rater 2: time: ES: 1,35 (?); p: 0,007; errors: ES 1,08 (?); p: 0,016 |
| | | | | reliability | ICC: total time: h: 0.85 (+), p: 0.96 (+); total errors: h: 0.87 (+) p: 0.88 (+) |
| | | | | reliability: measurement error | total time: h: 1.3s (?); p: 0.80 (?); total errors: h: 0.60e (?); p: 0.70e (?); AUC-time: 85% (95% CI 0.72–0.99; p: .001) (?); AUC-errors: 86% (95% CI, 0.72–0.99; p: .001) (?); score with best discriminative ability: 43.28s, 4 errors |
| **Ankle Inversion Discrimination Apparatus for Landing (AIDAL)** | Han [69] | reliability: n = 23 12–11 (52%-48%) h: 23.7y, SD: 2.3 p: 24.1y, SD: 2.7 comparison: n = 36 22–14 (61%-39%) h: 23.7y, SD: 2.2 p: 23.4y, SD: 2.4 | CAI, healthy | reliability: test-retest reliability | ICC: whole group: 0.763 (+), h: 0.701 (+), p: 0.804 (+) |
| | | | | validity: hypotheses testing for construct approach | AIDAL-CAIT: rho: 0.401 (?) |
| | | | | responsiveness | AUC: 0.756 (?), sensitivity: 0.733 (?), specificity: 0.800 (?), MDC: 0.04 (?) |
| *Gait* | | | | | |
| *Physical Activity Level* | | | | | |

*Note*:*participants*: total amount; male-female (%); mean age, SD.

abbreviations: AIDAL = Ankle Inversion Discrimination Apparatus for Landing; CAI = Chronic Ankle Instability; CAIT = Chronic Ankle Instability Tool; FAI = Functional Ankle Instability; FES = Effect Size; h = healthy; LLL = Lower Limb Length; MDC = Minimal Detectable Change; p = patient; SDC = Smallest detectable change; SEM = Standard Error of Measurement; TTB = Time-to-boundary; VAS = Visual Analog Scale

results of the criteria of good measurement property are added in brackets: +: sufficient; -: insufficient;?: indeterminate

table adapted from COSMIN recommendations

position and the use of Reverse Anterolateral Drawer Test were recommended. Studies investigating a population with more than four-week history of ankle sprain showed good MPs for the Cumberland Ankle Instability Tool as a PROM for the diagnosis of chronic and functional ankle instability and the Multiple Hop or Star Excursion Balance test for dynamic postural balance evaluation. Single studies investigated single tests on swelling, range of motion, strength, and static postural balance.

## General interpretation of results in context of other evidence

The basis for this review were the recommendations of the IAC´s COS summarized in the ROAST guideline. Consequently, the results of this literature search are now compared with the ROAST guideline.

Pain assessment is not only included in the assessment of acute cases in ROAST, but also in the decision on return to sport in the PAAS framework [70]. The authors of the ROAST guideline recommended the Numeric Analog Scale [71]. This instrument is reliable and valid for elderly or patients with low back pain or chronic pain [72]. It is likely MP data generated from other musculoskeletal conditions can be transferred to the ankle sprain population. Therefore, until further evidence is available the NAS should be used for pain evaluation.

Our data for ankle swelling assessment correspond to other studies reporting the Figure of Eight measurement tool to be reliable. Similar ICC values for reliability (ICC > 0.99) were found in a study in 29 acute ankle sprain subjects aged 18 to 59 years [73]. The study was excluded in this systematic review based on the inclusion criteria. Other studies that served as a reference in the ROAST guideline examined either a healthy population [74] or subjects after surgery following a malleolar fracture [75].

ROAST guidelines recommend the Weight Bearing Lunge test to assess ankle joint specific mobility. This consensus was based on studies with a relatively heterogeneous or even healthy population [76–78]. In the current review only one study investigated this test showing good responsiveness values [38]. No information was found for other MPs of this test for the target population. As mobility is also a recommended parameter for return to sport [70], this outcome should continue to be investigated. Alternative assessment techniques for ankle joint mobility were found to be reliable in healthy subjects. This includes, the use of a universal goniometer (inter-rater reliability: ICC: 0.76–0.87, intra-rater reliability: ICC: 0.85–0.91), and a smartphone goniometer (inter-rater reliability: ICC: 0.82–0.89, intra-rater reliability: ICC: 0.82–0.91) [79]. Future studies will determine if these results are applicable to patients with LAS pathology. It should also be assessed whether and to what extent the testing of mobility in LAS patients differs from other ankle pathologies.

For the determination of ankle strength, results of this present study showed good reliabilities of isokinetic strength measurement for ankle inversion and eversion. The authors of the ROAST guideline referred to a study on handheld dynamometer measurements in healthy subjects. It was reliable and valid in a heterogenous population. Therefore, the measurement of ankle strength using an (adapted) handheld dynamometer could still be an option [12, 80]. Corresponding studies for the evaluation of acute cases with LAS are currently lacking. This also applies to the isometric plantar flexion strength test which is considered another easy way of strength testing. In healthy subjects and a wide orthopedic population this test was reliable and valid [81]. However, as not 75 percent of the included population were diagnosed with LAS, the test was not eligible for inclusion in the present systematic review. A study on peroneal reaction time was excluded during full-text screening [82]. This test does not assess peak force values which are specific for strength testing. It is more related to ankle muscle functionality during specific movements but not to the ankle strength impairment defined within the ROAST guideline.

In qualitative analysis of the data of included studies on arthrokinematics for acute LAS, different results were found for the diagnostic accuracy of the ADT. This finding may be, at least partially, explained by differences in the time from injury to the time of examination across studies. An improvement in specificity and sensitivity during a delayed examination was also supported by Vuurberg and colleagues [83]. In other studies, the examination to determine diagnostic accuracy took place after more than one week. Additionally, the position of the examination can potentially influence test results. While Van Dijk performed the ADT in supine position [57, 58], the same test was performed in sitting position in other studies [59–61]. Moreover, the different values for diagnostic accuracy and reliability in both populations might be related to the subjectivity of the testing and different threshold values being applied by different investigators. Netterström-Wedin and colleagues recommended a cluster examination for the anterior talofibular ligament consisting of palpation and ADT [84]. This was due to high sensitivity of the palpation and high specificity of the ADT. Those results align with the findings of this study for physical examination possibly also due to the same reference studies. Due to the high specificity (100%, 99%) [60, 64] of ultrasonography, it is recommended for the identification of structural injuries following LAS. In general, it is noticeable that all the tests examined only look at one plane, especially the sagittal plane. The frontal plane and rotational component are neglected—as is the combination of several planes. Accordingly, a more specific answer to the recommended tests or confirmation of this recommendation is not possible. This systematic review found no study which examined the Posterior Talar Glide Test in LAS patients, which was recommended by the ROAST guideline [85].

Conclusions on PROMs of the current literature were inconsistent. The present study showed evidence for individual MPs for some questionnaires. Especially in the acute phase,

PROMs were only investigated in single studies. The Ankle Function Scale and the Sport Ankle Rating System might be helpful for acute ankle sprain evaluation, but further evidence is needed. In contrast, CAIT and its MPs were comprehensively investigated in several studies for individuals with CAI. Both reliability and validity values were found for the Quick-FAAM but not for the FAAM questionnaire [27–29]. These results differ from the recommendations of the FADI and FAAM questionnaires within the ROAST guideline. The IAC supported their recommendation on a study [34] also included in this paper with indeterminate results in the area of hypotheses testing for construct validity according to COSMIN guidelines. Additionally, the ROAST guideline referenced a development study of the FADI [86]. Moreover, another systematic review from 2021 showed once again different results. Hansen and colleagues evaluated content validity and MPs in relevant PROMs for ankle instability assessment. They argued that patient involvement during the development phase of PROMs was crucial for the content validity. Only three of the 23 investigated PROMs in this study included patients during this phase (Cumberland Ankle Instability Tool, Lower Extremity Functional Scale, Foot and Ankle Ability Measure). Using the quality assessment system by Hansen and colleagues only the FAAM questionnaire demonstrated adequate construct validity [71]. Therefore, they recommended to use the FAAM questionnaire for assessing ankle instability [87]. However, since health-related quality of life differences between patients with chronic ankle instabilities and healthy persons could be expected, PROMS should be able to detect such difference in quality of life. Yet, this was currently only inadequately fulfilled [88]. Hansen's systematic review also showed overall insufficient evidence for construct validity of the CAIT questionnaire. Future studies should therefore comprehensively examine the most important PROMs in several MPs in people with ankle instabilities to be able to make more comprehensive recommendations for or against the use of a specific PROM. In the meantime, it is advisable that the questionnaires FAAM, FADI or CAIT should be used.

The referenced studies in the ROAST guideline only used Balance Error Scoring System and Foot Lift Test to investigate whether postural balance was reduced in a population with functional ankle instability. However, on the one hand the MPs of the tests for detection of static postural balance deficits were not specifically investigated [89, 90]. Based on the included literature of this review, no recommendations can be made for or against the examination in general or the use of the two tests. Both tests were assessed for their validity and responsiveness but showed indeterminate results based on the COSMIN criteria for both MPs [42]. This should be investigated in future studies.

On the other hand, dynamic postural control is not only recommended in the acute setting but also for testing before a return to sport decision [70]. Similar results for the Star Excursion Balance Test were found in this present study, as recommended in the ROAST guideline. This was also in line with the literature on the test or the slightly modified Y-balance test in general, regardless of the population [91, 92]. Besides the use of the recommended Star Excursion Balance Test adequate evidence was found for the Multiple Hop Test in this present systematic review. Both tests are practicable, inexpensive and relatively quick to perform. Future research should evaluate whether there are differences between the two tests in terms of assessed parameter or in which situations one of the two tests would be more suitable.

The experts who developed the ROAST guideline recommend the visual assessment of antalgic gait. However, they did not reference any study to it. The lack of studies investigating tests for gait assessment was also found in this present review. The extent to which gait can be objectively and validly assessed should be the focus of future studies. A study investigating gait initiation using a treadmill with integrated force plates was excluded during full-text screening [93]. Authors of the ROAST recommendations stated that "ankle sprain injury recurrence during gait is likely due to inappropriate positioning of the lower extremity joints in the loading-

unloading transitions between stance and swing" [12]. Hartley and colleagues'investigation on gait initiation does not fall under this definition and was therefore not considered in this systematic review.

No statement can currently be made about the physical activity level due to a lack of studies. A study on knee ligament injuries was used as a reference in the ROAST guideline [94]. The transferability to LAS is questionable.

## Limitations of evidence included in the review

The evidence for all outcomes remains limited in this systematic review due to the small number of studies per test and investigated MPs. In addition, apart from six studies on arthrokinematics and two on PROMs, only the population at least four weeks after a LAS was examined. Studies in this population are difficult to carry out, as a study entry would have to be considered immediately after a possible consultation in the emergency center, with the general practitioner or orthopedist. Also, due to the inflammatory response, pain and swelling, the tests under evaluation are often not as sensitive as the arthrokinematics test studies of van Dijk recognized [57, 58]. This may explain the better diagnostic accuracy values of a delayed examination with then decreasing symptoms. However, the acute assessment of LAS is important to classify the extent of the injury and to guide rehabilitation [95]. Additionally, studies on responsiveness of tests are lacking. These are essential for the clinically active professionals to be able to make a statement about the test.

It should also be noted that the tests of included arthrokinematic studies primarily serve to diagnose injuries to the lateral ligamentous apparatus of the ankle joint. The ROAST guideline included the posterior talar glide test, a test to assess posterior glide of the talus within the talocrural joint [12]. Studies for the latter area could not be found within the framework of the systematic review. One of the included studies evaluated CAI with a load displacement ratio using a digital arthrometer. This study stated intraclass correlation coefficient values for testretest (ICC 0.897 to 0.963) as well as for inter-tester reliability (ICC 0.949). As it was not mentioned as an outcome of the study and detailed the methodology of data collection for the reliability of this test, however, this outcome was not included in this review [66].

The heterogeneity across the included studies made pooling the data not plausible. For example, several studies were found in the areas of arthrokinematics, PROMs or dynamic postural balance. Since at least one of the factors of the study population characteristics, the test itself or the MP differed, the results could not be pooled. As a result, the GRADE approach recommended by COSMIN could also not be performed. This would have been helpful to formulate generalizable statements for or against a certain test.

Dividing the population into two subgroups primarily based on time since first injury is questionable. During the rehabilitation a rapid change in self-reported function, ankle range of motion and pain is observed in the first four weeks. However, patients with this diagnosis are seldom functionally and structurally completely recovered at four weeks post injury [96]. Symptoms do persist for a longer period. Similar results of symptom persistence were found in another study. Participants returned to play after on average with 12.70 days. They were not free of structural or functional deficits at the time of return to sport [97]. Moreover, there is no clear definition of when one can speak of a successful return to sport. Therefore, the division in two subgroups based on time since injury is a limitation of this systematic review.

The COSMIN guideline was easy to use for case-control and cross-sectional studies MPs [15]. It was more difficult to use in studies that determine diagnostic accuracy. No guideline existed for this determination and its interpretation possibly also due to the limitation of the guideline to PROMS. Additionally, a lot of MPs especially for hypotheses testing and

responsiveness were rated with an indeterminate result based on the COSMIN criteria for good MPs. This is primarily due to no hypothesis definition of the research team. Maybe a more accurate definition of criteria for those MPs would be advisable.

## Strengths and limitations of the review processes used

The present review was developed according to the COSMIN criteria for systematic reviews for PROMs. Together with the independent screening, evaluation, and data extraction by two authors at each stage of study development risk for publication bias may be low. In addition, most studies were assessed with a low risk of bias in the three assessment tools (COSMIN, adapted-COSMIN, QUADAS-2). However, the COSMIN criteria were not developed for clinical trials. Such list is currently not available.

Besides of these strengths, some limitations need to be recognized. The selection and reporting bias seems to be relatively small, as two authors carried out each step independently of each other and a third author was consulted in case of discrepancy. When focusing on the criteria for good MPs, it becomes clear that outcomes for measurement error, hypothesis testing and responsiveness were mostly assessed with an indeterminate result. This finding may have resulted because research teams of the included studies had not defined a clear hypothesis for these criteria. In the measurement error category, not all values (smallest detectable change, minimally important change or limits of agreement) were defined in the respective studies. This meant that in the end in most of the cases no clear decisions could be made for or against the use of specific assessments.

## Implications of the results for practice, policy and future research

Recommendations for testing a LAS begin after differential diagnosis and exclusion of fracture or syndesmosis involvement using the Ottawa Ankle Rules or specific syndesmosis testing [98, 99]. The current literature does not allow exclusion of specific outcomes from the ROAST guideline. Thus, pain, ankle mobility, gait and physical activity level should be assessed with the tests recommended by the expert group until new or more comprehensive literature is available. The Figure of Eight test for swelling assessment is still recommended. Arthrokinematic tests should continue to be performed with ADT, RADT, TTT and the Posterior Talar Glide test as recommended by the ROAST guideline. Especially analyzing strength with an isokinetic dynamometer is a relatively unusual piece of equipment in clinical setting such as the private physiotherapy practice. Measurement characteristics of cost-effective methods that are more applicable to clinical practice, such as Hand-Held Dynamometer testing, should be explored in the future. Future studies should also assess different setups and positions of testing. In contrast, such isokinetic dynamometer is more common in huge rehabilitation facilities or especially in research. Accordingly, testing should be left primarily to those institutions for the time being.

If time is limited for patient evaluation possibly the examination of static postural balance could be excluded until new evidence exists. Star Excursion Balance Test and possibly the Multiple Hop Test can be used for dynamic postural control.

Future research should especially focus on range of motion and strength testing. Arthrokinematic tests in frontal, rotational plane or even in more than one plane could be useful too as ankle movement is relatively complex and seldom in only one plane. Studies comparing the MPs and outcomes of the Star Excursion Balance Test and Multiple Hop Test should also be conducted. This can also be important for monitoring progress during rehabilitation [100]. The same applies to the CAIT, FAAM, Quick-FAAM and FADI questionnaires. Further studies are needed for all other outcomes especially focusing on the acute assessment.

## Conclusion

This systematic review showed important clinical implications. The strongest evidence relates to clinical tests diagnosing acute injury to the ATFL injury, and PROMs for assessing functional instability in a CAI population. Clinicians are advised to use a delayed (more than five days post injury) ADT in supine position, and Reverse Anterolateral Drawer Test in acute assessment. The Cumberland Ankle Instability Tool is a valid and reliable PROM for individuals with chronic or functional ankle instability. Dynamic postural balance assessment using Multiple Hop and Star Excursion Balance Test was also recommended. Future studies in this should prioritize assessment in the acute stages after sprain, and strength testing. More prospective studies are needed to determine the responsiveness of tests used to assess all impairments.

## Supporting information

**S1 Checklist. PRISMA 2020 checklist.**
(DOCX)

**S1 Table. Search strategies.** Search terms for all six databases.
(DOCX)

**S2 Table. List of excluded studies during full-text screening.**
(DOCX)

## Author Contributions

**Conceptualization:** Alexander Philipp Schurz, Jente Wagemans, Chris Bleakley.

**Data curation:** Alexander Philipp Schurz.

**Formal analysis:** Alexander Philipp Schurz, Jente Wagemans, Chris Bleakley.

**Investigation:** Alexander Philipp Schurz, Jente Wagemans.

**Methodology:** Alexander Philipp Schurz, Jente Wagemans, Chris Bleakley.

**Supervision:** Kevin Kuppens, Dirk Vissers, Jan Taeymans.

**Visualization:** Alexander Philipp Schurz.

**Writing – original draft:** Alexander Philipp Schurz.

**Writing – review & editing:** Jente Wagemans, Chris Bleakley, Kevin Kuppens, Dirk Vissers, Jan Taeymans.

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
