## [Decision Letter · Decision Letter 0]

1 Dec 2022

PONE-D-22-30101Impairment-Based Assessments for Patients With Lateral Ankle Sprain: A Systematic Review of Measurement PropertiesPLOS ONE

Dear Dr. Schurz,

Thank you for submitting your manuscript to PLOS ONE. After careful consideration, we feel that it has merit but does not fully meet PLOS ONE’s publication criteria as it currently stands. Therefore, we invite you to submit a revised version of the manuscript that addresses the points raised during the review process.

ACADEMIC EDITOR:Dear Author,There are still some issues that need correction. Please attend to all the reviewers' comments and make the necessary changes.==============================

We look forward to receiving your revised manuscript.

Kind regards,

Zulkarnain Jaafar

Academic Editor

PLOS ONE

Journal Requirements:

"No"

"No"

Reviewers' comments:

Reviewer's Responses to Questions

**Comments to the Author**

1. Is the manuscript technically sound, and do the data support the conclusions?

Reviewer #1: Yes

Reviewer #2: Yes

2. Has the statistical analysis been performed appropriately and rigorously? 

Reviewer #1: Yes

Reviewer #2: Yes

3. Have the authors made all data underlying the findings in their manuscript fully available?

Reviewer #1: Yes

Reviewer #2: Yes

4. Is the manuscript presented in an intelligible fashion and written in standard English?

Reviewer #1: Yes

Reviewer #2: Yes

5. Review Comments to the Author

Reviewer #1: Thank you for the opportunity to review "Impairment-Based Assessments for Patients With Lateral Ankle Sprain: A Systematic Review of Measurement Properties "

The paper provides a thorough review of the measurement properties of several physical measures and PROMs appropriate for the care of patients with ankle injuries. I have few comments.

Line 102 - should read "data were"

Line 202 - I do not agree with this statement. It seems unclear as to the timing of the assessment of the subjects in this study. Likely that some / most were < 4 wks post injury resulting in swelling given that the sample was drawn from a college campus.

line 248 - is this 53-93% or 5.3 to 93%? if the latter simply state 5-93% for clarity

line 397 - The use of ultrasound is recommended ? please clarify as the sentence does not read well

459 - Perhaps "Due to the inflammatory response, pain and swelling, the test .... " would better convey your intention

Reviewer #2: Review

Many thanks to the authors for having presented a so interesting systematic review about “Impairment-Based Assessments for Patients with Lateral Ankle Sprain: A Systematic Review of Measurement Properties”.Before resubmitting the revision version of the article, please read the editorial rules carefully, and check other editorial aspects. The language is partially good, but the manuscript needs to be corrected by a person of English mother tongue.

Abstract

The abstract is partially good, please mention the type of the study, research timing and study settings.

This section is partially well structured and contains the main information of the study.

Key words

Please provide them in alphabetic order and add a couple of new key words.

Background

The introduction explains well the scientific premises of the study, and why this topic was chosen. However, it would be necessary to better explain the hypotheses that must be verified or denied by the study.

Please, provide more anatomical deteails of both lateral and medial compartment of the ankle joint, quoting:

• The undefined anatomical variations of the deltoid ligament bundles: A cadaveric study. 10.11138/mltj/2018.8.2.163

Methods

Please add the exact period within which study selection was conducted. Please describe more fully what you have done to avoid or address possible study bias.

Statistical analysis

Please describe the statistical analysis methos in more detail way: who performed the analysis: an independent statistician or the same authors?

Results

The results presented are quite complete, reflecting the MM section.

Please displayed the results in a readable fashion.

Discussion

The length and content of the discussion communicates the main information of the paper. However, the results should be better discussed with those presented in the most recent literature, quoting:

• Is Kinesio Taping Effective for Sport Performance and Ankle Function of Athletes with Chronic Ankle Instability (CAI)? A Systematic Review and Meta-Analysis. Medicina (Kaunas). 2022 Apr 29;58(5):620. doi: 10.3390/medicina58050620.

Please, better describe any potential bias and weaknesses of the study.

Conclusion

The conclusions only reflect and refer to the results of the study and it is justified by the results and the methods.

References

The references are not up to date. Hence, delate those before 2010 if not essential, replacing them with newer ones. Please correct the references making them all the same in style according to editorial rules.

Tables and Figures

The number and quality of tables and figures are appropriate to transmit the main information of the paper.

6. PLOS authors have the option to publish the peer review history of their article (what does this mean?). If published, this will include your full peer review and any attached files.

Reviewer #1: **Yes: **Craig R Denegar

Reviewer #2: No

---

## [Author Response · Author response to Decision Letter 0]

9 Dec 2022

Dear Reviewers,

Thank you for your feedback on our systematic review. 

We have taken all of them into account in the revised version. 

The layout of the manuscript including tables and figures and the bibliography have been adapted. Aspects that were still desired have been specified. 

The second reviewer requested the referencing of two studies in the background and discussion sections. Since we want to focus on the resulting impairments and less on the anatomical conditions of the ankle joint in the introduction, we did not include the cadaver study. In addition, the anatomical conditions of the deltoid ligament on the medial side of the ankle are primarily described here. The 2nd study should be cited in the discussion section. However, since this study primarily considers the interventional effects of kinesiotaping and our study primarily examines the testing of the impairments, we have not included this study either.

We are open to any feedback.

Thank you for your helpful feedback!

With kind regards,

Alexander Schurz and colleagues

---

## [Decision Letter · Decision Letter 1]

20 Dec 2022

PONE-D-22-30101R1Impairment-Based Assessments for Patients With Lateral Ankle Sprain: A Systematic Review of Measurement PropertiesPLOS ONE

Dear Dr. Schurz,

Thank you for submitting your manuscript to PLOS ONE. After careful consideration, we feel that it has merit but does not fully meet PLOS ONE’s publication criteria as it currently stands. Therefore, we invite you to submit a revised version of the manuscript that addresses the points raised during the review process.

ACADEMIC EDITOR: Dear Author,Please attend to all the points that need to be revised as raised by the reviewer/s.The decision of this manuscript is justified based on PLOS ONE’s publication criteria and not its on novelty or perceived impact.

We look forward to receiving your revised manuscript.

Kind regards,

Zulkarnain Jaafar

Academic Editor

PLOS ONE

Reviewers' comments:

Reviewer's Responses to Questions

**Comments to the Author**

1. If the authors have adequately addressed your comments raised in a previous round of review and you feel that this manuscript is now acceptable for publication, you may indicate that here to bypass the “Comments to the Author” section, enter your conflict of interest statement in the “Confidential to Editor” section, and submit your "Accept" recommendation.

Reviewer #1: All comments have been addressed

Reviewer #2: All comments have been addressed

2. Is the manuscript technically sound, and do the data support the conclusions?

Reviewer #1: Yes

Reviewer #2: Yes

3. Has the statistical analysis been performed appropriately and rigorously? 

Reviewer #1: Yes

Reviewer #2: Yes

4. Have the authors made all data underlying the findings in their manuscript fully available?

Reviewer #1: Yes

Reviewer #2: Yes

5. Is the manuscript presented in an intelligible fashion and written in standard English?

Reviewer #1: Yes

Reviewer #2: Yes

6. Review Comments to the Author

Reviewer #1: (No Response)

Reviewer #2: Please, quote the articles, related with the study topic, as suggested in the previous revision of the original manuscript. Thank you!

7. PLOS authors have the option to publish the peer review history of their article (what does this mean?). If published, this will include your full peer review and any attached files.

Reviewer #1: **Yes: **Craig R Denegar

Reviewer #2: No

---

## [Author Response · Author response to Decision Letter 1]

21 Dec 2022

Dear Second Reviewer,

I have now added the 2 quotations as wished. These are added in the background and discussion section.

Thank you!

Best wishes,

Alexander Schurz

---

## [Editor Report · Decision Letter 2]

28 Dec 2022

Impairment-Based Assessments for Patients With Lateral Ankle Sprain: A Systematic Review of Measurement Properties

PONE-D-22-30101R2

Dear Dr. Schurz,

We’re pleased to inform you that your manuscript has been judged scientifically suitable for publication and will be formally accepted for publication once it meets all outstanding technical requirements.

Kind regards,

Zulkarnain Jaafar

Academic Editor

PLOS ONE
---

## [Editor Report · Acceptance letter]

3 Jan 2023

PONE-D-22-30101R2 

Impairment-based assessments for patients with lateral ankle sprain: A systematic review of measurement properties 

Dear Dr. Schurz:

I'm pleased to inform you that your manuscript has been deemed suitable for publication in PLOS ONE. Congratulations! Your manuscript is now with our production department. 

Kind regards, 

on behalf of

Dr. Zulkarnain Jaafar 

Academic Editor

PLOS ONE